# Sarc-Graph: Automated segmentation, tracking, and analysis of sarcomeres in hiPSC-derived cardiomyocytes

**Bill Zhao**[1], **Kehan Zhang**[2,3], **Christopher S. Chen**[2,3], **Emma Lejeune**[1]*

**1** Department of Mechanical Engineering, Boston University, Boston, Massachusetts, United States of America, **2** Department of Biomedical Engineering, Boston University, Boston, Massachusetts, United States of America, **3** The Wyss Institute for Biologically Inspired Engineering, Harvard University, Cambridge, Massachusetts, United States of America

* elejeune@bu.edu

**Data Availability Statement:** The data underlying the results presented in the study are available from https://github.com/elejeune11/Sarc-Graph. In S1, S2 and S4 Texts we compare our work to other

## Abstract

A better fundamental understanding of human induced pluripotent stem cell-derived cardiomyocytes (hiPSC-CMs) has the potential to advance applications ranging from drug discovery to cardiac repair. Automated quantitative analysis of beating hiPSC-CMs is an important and fast developing component of the hiPSC-CM research pipeline. Here we introduce "Sarc-Graph," a computational framework to segment, track, and analyze sarcomeres in fluorescently tagged hiPSC-CMs. Our framework includes functions to segment z-discs and sarcomeres, track z-discs and sarcomeres in beating cells, and perform automated spatio-temporal analysis and data visualization. In addition to reporting good performance for sarcomere segmentation and tracking with little to no parameter tuning and a short runtime, we introduce two novel analysis approaches. First, we construct spatial graphs where z-discs correspond to nodes and sarcomeres correspond to edges. This makes measuring the network distance between each sarcomere (i.e., the number of connecting sarcomeres separating each sarcomere pair) straightforward. Second, we treat tracked and segmented components as fiducial markers and use them to compute the approximate deformation gradient of the entire tracked population. This represents a new quantitative descriptor of hiPSC-CM function. We showcase and validate our approach with both synthetic and experimental movies of beating hiPSC-CMs. By publishing Sarc-Graph, we aim to make automated quantitative analysis of hiPSC-CM behavior more accessible to the broader research community.

## Author summary

Heart disease is the leading cause of death worldwide. Because of this, many researchers are studying heart cells in the lab and trying to create artificial heart tissue. Recently, there has been a growing focus on human induced pluripotent stem cell-derived cardiomyocytes (hiPSC-CMs). These are cells that are safely sampled from living humans, for example from the blood or skin, that are then transformed into human heart muscle cells. One

PREVIOUSLY PUBLISHED frameworks. In S1 Text, the SOTA code is implemented in MATLAB and available from https://github.com/saucermanlab/SarcOrgTextureAnalysis, and the data is available from https://figshare.com/articles/dataset/Sutcliffe_et_al_Scientific_Reports_2018_images/11390022. In S2 Text, the data is available from http://diseasebiophysics.seas.harvard.edu/publications. In S4 Text, the SarcTrack code is implemented in MATLAB and available from https://github.com/HMS-IDAC/SarcTrack.

**Funding:** This work was supported by the CELL-MET Engineering Research Center National Science Foundation ECC-1647837 (CSC is a co-PI, EL is an affiliated faculty). See: https://www.bu.edu/cell-met/ and https://nsf.gov/awardsearch/showAward?AWD_ID=1647837 KZ acknowledges fellowship support from the American Heart Association, award number 17PRE33660967. See: https://www.heart.org/. EL acknowledges the Boston University Hariri Junior Faculty Fellowship, and the Boston University David R. Dalton Career Development Professorship. The funders had no role in study design, data collection and analysis, decision to publish, or preparation of the manuscript.

**Competing interests:** The authors have declared that no competing interests exist.

active research goal is to use these cells to repair the damaged heart. Another active research goal is to test new drugs on these cells before testing them in animals and humans. However, one major challenge is that hiPSC-CMs often have an irregular internal structure that is difficult to analyze. At present, their behavior is far from fully understood. To address this, we have created software to automatically analyze movies of beating hiPSC-CMs. With our software, it is possible to quantify properties such as the amount and direction of beating cell contraction, and the variation in behavior across different parts of each cell. These tools will enable further quantitative analysis of hiPSC-CMs. With these tools, it will be easier to understand, control, and optimize artificial heart tissue created with hiPSC-CMs, and quantify the effects of drugs on hiPSC-CM behavior.

This is a *PLOS Computational Biology* Software paper.

## Introduction

Quantitative analysis of movies of beating cardiomyocytes is a compelling approach for connecting cell morphology to dynamic cell function [1, 2]. In particular, connecting structure and function is a crucial step towards a better fundamental understanding of human induced pluripotent stem cell-derived cardiomyocytes (hiPSC-CMs) [3, 4]. In stark contrast to sarcomere chains in mature cardiomyocytes which have a highly ordered regular structure [5], sarcomere chains in hiPSC-CMs are typically immature and disordered [6]. This irregular structure, combined with large variation between cells, makes developing tools for quantitative analysis both more difficult and more pressing [7]. Robust quantitative analysis frameworks for analyzing beating hiPSC-CMs will help enable technological advances in drug discovery, genetic cardiac disease, and cardiac repair [8–10]. In this work, we aim to make automated quantitative analysis of hiPSC-CM contractile behavior more accessible to the broader research community.

At present, there are multiple different methods available in the literature for analyzing the morphology of both mature cardiomyocytes and hiPSC-CMs. However, most of these methods are designed for still images (often of fixed cells) rather than dynamic movies of beating cells. Recently Morris et al. 2020 [11] developed the structural assay "ZlineDetection," a computational tool to segment and analyze sarcomeric z-discs. Sutcliffe et al. 2018 [12] developed "SarcOmere Texture Analysis" (SOTA), a computational tool for quantifying sarcomere structure using Haralick texture features. And, Pasqualini et al. 2015 [6] defined 11 metrics for describing sarcomere structure (sarcomere length, total energy, sarcomeric energy, sarcomeric packing density, orientational order parameter (OOP), sarcomeric OOP, nonsarcomeric OOP, Z-disc relative presence, weighted OOP, coverage quality control, and coherency quality control). In the second two examples, filter operations are applied to each image and single quantities of interest are extracted for the entire field of view [6, 12]. Though it is possible to analyze a movie as a sequence of images and compare average changes, none of these tools have built in functionality for tracking individual sarcomere motion and length changes between image frames.

In well aligned and synchronously beating cardiomyocytes, Fast Fourier Transforms (FFT) can be used to determine sarcomere length in a chain of sarcomeres with a (often manually)

defined axis [2]. We note that this is readily available as "SarCoptiM," an ImageJ plugin [2, 13]. However, because sarcomere chains in hiPSC-CMs are often not in alignment and individual sarcomeres often do not deform in sync, these methods are not necessarily suitable for robust automated analysis of hiPSC-CM movies. Recently, Ribeiro et al. 2017 [8] developed a multi-imaging (brightfield, fluorescent beads, and an actin stain) based assay to quantify the mechanical contractile output of hiPSC-CMs. For this method, cells are seeded onto a micropatterned deformable polyacrylamide substrates containing fluorescent microbeads. This setup makes it possible to not only image dynamic cell deformation, but also measure the contractile forces that the cell is exerting on the substrate. Though the authors explored multiple different approaches for calculating sarcomere length, sarcomeres were not explicitly tracked between frames. Finally, Toepfer et al. 2019 [10] developed "SarcTrack," a tool for segmentation and tracking of individual sarcomeres in movies of beating hiPSC-CMs. With this software, sarcomeres are fitted with Morlet wavelets, and assumed to contract following a periodic sawtooth function. Because individual sarcomeres are tracked, it is possible to report quantities of interest such as sarcomere contraction and relaxation time and sarcomere shortening on an individual sarcomere basis.

Movies of fluorescently labeled hiPSC-CMs contracting are incredibly information rich [10]. Fig 1A shows a hiPSC-CM with fluorescently labeled z-discs. Critically, with dynamic data it is possible to not only measure structure but also measure aspects of how the structural components functionally interact [4, 14–17]. The focus of this work is on extracting structural and functional information from movies of beating hiPSC-CMs with fluorescently labeled z-discs. Here, we provide analysis tools beyond the scope of SarcTrack [10], the current state of the art software for sarcomere segmentation and tracking. In addition, we provide an alternative approach to both sarcomere segmentation and tracking that may be more effective in some scenarios. Notably, our code has the practical benefits of requiring little to no manual parameter tuning and implementation in the open source Python programming language.

To extract biologically relevant quantitative information from these movies, we focus primarily on segmenting, tracking, and analyzing individual sarcomeres. This would not be possible with alternative approaches where one average metric is computed per image. Each movie will typically have over 100 frames, and each frame will contain 100s of sarcomeres. Therefore, it is infeasible to perform segmentation and tracking by hand at scale. In the first component

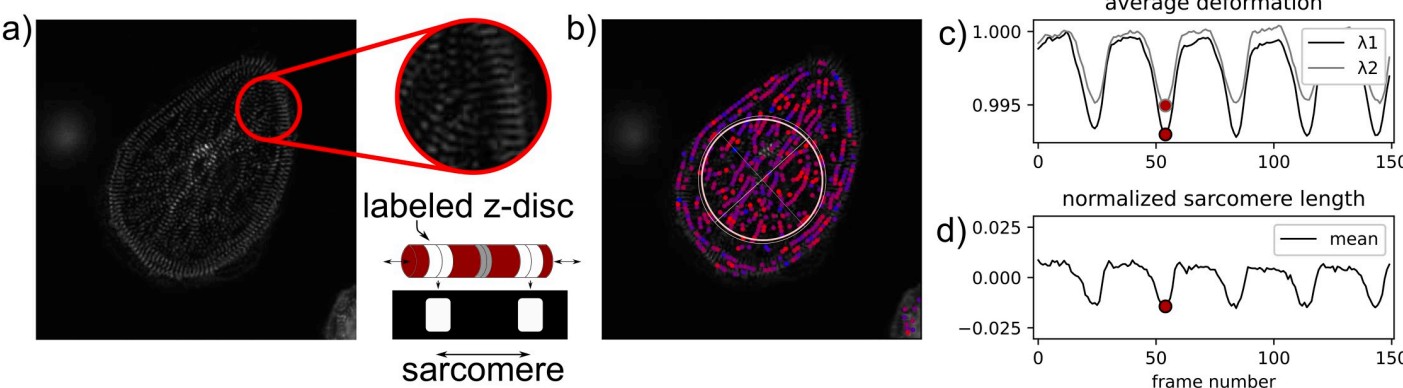

**Fig 1.** Analyzing movies of contracting human induced pluripotent stem cell-derived cardiomyocytes (hiPSC-CMs): (a) Still image of a beating hiPSC-CM movie. As per the schematic, z-discs, which define the ends of an $\approx 2 \, \mu m$ sarcomere, are fluorescently labeled; (b) Movie still with a schematic illustration of deformation gradient $\mathbf{F}_{avg}$ and tracked sarcomeres overlaid, sarcomere color corresponds to contraction level with red indicating the highest level of contraction; (c) Magnitudes of average principal stretches ($\lambda_1, \lambda_2$) computed from $\mathbf{F}_{avg}$ with respect to the movie frame number; (d) Average normalized sarcomere length with respect to the movie frame number. We note that the definitions of $\mathbf{F}_{avg}$, $\lambda_1$, and $\lambda_2$ are introduced in this text.

of our computational framework, we introduce a procedure for automated sarcomere segmentation and tracking that explicitly captures individual sarcomeres. Example results from segmentation and tracking are shown in Fig 1B.

Building on effective segmentation and tracking, the second contribution of our computational framework is tools for automated analysis of hiPSC-CM contractile behavior. To accomplish this, we first draw inspiration from recent literature on defining summary metrics for describing cardiomyocyte structure and cytoskeletal structure [11, 18] and mechanical deformation in multiple cell types [19, 20]. Specifically, we treat the tracked sarcomeres as fiducial markers and quantify the principal directions and magnitudes of average cell contraction for each movie frame. An example of applying this approach is shown in Fig 1C, with Fig 1D for comparison. Then, we move beyond average metrics and extend our framework to readily perform spatiotemporal analysis of sub-cellular sarcomere contraction. Specifically, we treat the structurally disordered and complex hiPSC-CMs as spatial graphs where z-discs are represented as nodes and sarcomeres are represented as edges. With a spatial graph defined, performing spatial statistics based analysis is straightforward. For example, it is possible to examine the correlation between individual sarcomere contraction time series using both Euclidean and network distances.

In conjunction with segmentation, tracking, and analysis, our computational framework—Sarc-Graph—also contains multiple visualization tools. The remainder of this paper is focused on describing key components of the framework. We note that all code is available free and open source on GitHub: https://github.com/elejeune11/Sarc-Graph. The code is designed in a modular way such that making major alterations to one component or performing only a subset of the segmentation, tracking, and analysis steps should be straightforward when appropriate. Looking forward, we anticipate that this work is a starting point for a major effort in quantification and statistical analysis of hiPSC-CM contractile behavior.

## Materials and methods

### Data

In general, it is infeasible to perform segmentation and tracking by hand at scale for movies of hiPSC-CMs with fluorescently labeled z-discs. To address the lack of "ground truth" data we invest significant effort in realistic synthetic data generation to validate the segmentation and tracking components of our computational framework. In addition, we show the results of applying our framework to a selection of real experimental movies.

**Synthetic data for software validation.** In Fig 2, we show the critical components of the synthetic data generation pipeline. The first step is illustrated in Fig 2A where we show an example of generating the three-dimensional (3D) skeleton of z-disc and sarcomere geometry for each movie frame. First, a baseline geometry is defined. Then, the baseline geometry is deformed in space such that each sarcomere is associated with a ground truth function for contraction (shown here as fractional change in length) with respect to time. By starting with this 3D skeleton geometry, it is straightforward to include examples with inhomogeneous contraction, out of plane orientation and deformation, and sarcomere chains that appear to intersect when projected into two-dimensional (2D) space.

Given the 3D skeleton geometry in Fig 2A, we then render each frame as illustrated in Fig 2B–2D. First, each z-disc is treated as an oriented cylinder with an associated height and radius. We note that with this approach it is straightforward to include multiple z-disc sizes in the same image. Then, the domain is converted into a voxel array such that the resolution is representative of the experimental images of interest. The voxel array, a 3D matrix, is then sliced around the simulated focal plane and projected into 2D. Both image noise and blur can

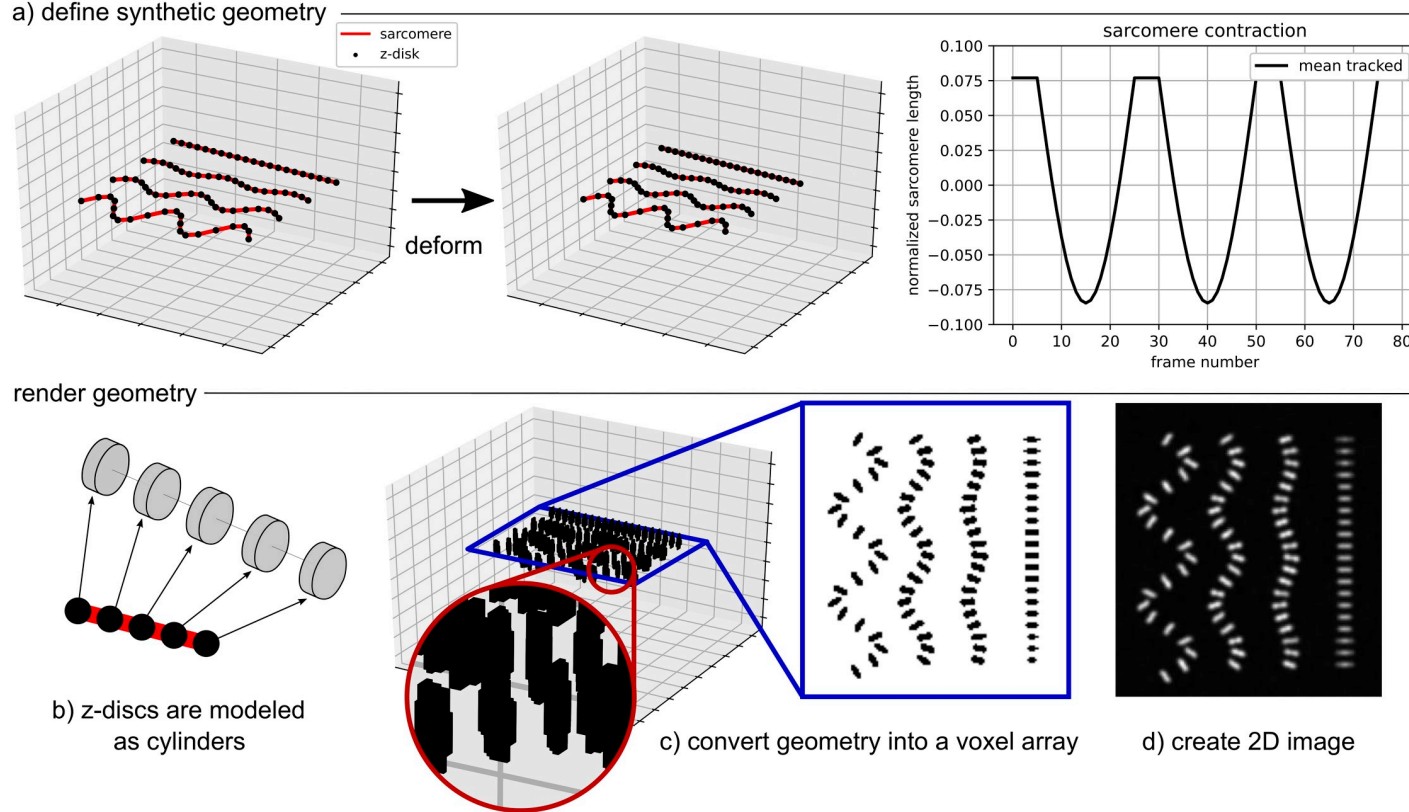

**Fig 2.** Creating synthetic data to test the image analysis code: (a) Define the geometry of z-discs (points) and sarcomeres (line segments) in three-dimensional (3D) space and prescribe the ground truth deformation for each sarcomere; (b) Approximate each z-disc as an oriented cylinder in 3D space; (c) Convert the 3D geometry into a voxel array with higher resolution in the x and y directions than the z direction; (d) Slice and project the voxel array to obtain a two-dimensional (2D) image. Image noise and blur can be added to the voxel array, 2D image, or both.

be added at the voxel array step, 2D image step, or both. In the representative examples shown here, we add Gaussian blur to the voxel array [21] and Perlin noise to the 2D image [22]. We note that the entire synthetic data generation pipeline is available on GitHub. All code is written in Python with heavy use of the numpy and matplotlib packages [23, 24].

**Experimental data for software demonstration.** The first two examples of experimental data included with this paper (E1 and E2) were previously made available in a publication by Toepfer et al. [10]. The protocol to introduce green fluorescent protein (GFP) onto z-disc protein titin (TTN-GFP) in hiPSC-CMs is reported in [26]. Video imaging was conducted on small clusters of hiPSC-CMs using a 100X objective of a fluorescent microscope with a minimum acquisition rate of 30 frames per second. The three additional examples included with this publication were selected to showcase the general applicability of Sarc-Graph. Cells in E3, E4, and E5 were fluorescently labeled using lentiviral constructs lenti-GFP-actinin-2 or lenti-mApple-actinin-2 as described in [25], and were electrically stimulated at 1Hz using the IonOptix C-Pace EP Culture Pacer (IonOptix) during imaging. Time-lapse videos of cell contraction were acquired at 30 frames per second using a 40x objective on a Nikon Eclipse Ti (Nikon Instruments, Inc.) with an Evolve EMCCD Camera (Photometrics) or on a Zeiss Axiovert 200M inverted spinning disk microscope with an ORCA-100 Camera (Hamamatsu), equipped with a temperature and CO2 equilibrated environmental chamber. We note that substantial additional detail on the experimental protocol has been provided in previous

**Table 1. Summary of experimental examples included in this paper (E1, E2, E3, E4, and E5).** We note that Examples E1 and E2 have already been published and made publicly available at https://github.com/HMS-IDAC/SarcTrack [10].

|  | key information on the experimental data | key Sarc-Graph analysis challenges | additional info |
|---|---|---|---|
| E1 | hiPSC-CMs cultured on a fibronectin coated glass substrate, vertically aligned fibers. | large deformation | [10] |
| E2 | hiPSC-CMs cultured on a fibronectin coated glass substrate, tangentially aligned fibers. | large deformation | [10] |
| E3 | Single paced hiPSC-CM constrained on a micropatterned island (2000 $\mu m^2$) with fibronectin coating on top of soft polyacrylamide hydrogel (7.9 kPa). The method for making the hydrogel is described in [25]. | low spatial resolution, large deformation | [25, 26] |
| E4 | Monolayer of paced hiPSC-CMs cultured on a fibronectin-coated glass substrate. | substantial background signal | [25, 26] |
| E5 | Single paced hiPSC-CMs cultured on a fibronectin-coated glass substrate. | low spatial resolution, small deformation | [25, 26] |

literature [25, 26]. Key details are summarized in Table 1. Though not included as examples with this publication, our computational framework has also been tested on additional beating hiPSC-CM movies obtained by different researchers with subtly different experimental set ups. The examples E3, E4, and E5 were deliberately chosen as diverse examples to challenge the performance of Sarc-Graph for multiple setups. We also note that for all examples shown in the main body of this paper, both experimental and synthetic, our framework does not require any parameter tuning. All input parameters are identical across all cases. To complement these data, we also analyze previously released data from other groups [6, 12] in the S1 and S2 Texts. Though these examples are of images rather than movies, they provide an additional opportunity to showcase the capabilities of Sarc-Graph on image segmentation under more diverse conditions and compare the performance of Sarc-Graph to other already established approaches. Further details are given in the S1 and S2 Texts.

## Code

Sarc-Graph is divided into seven modular components: `file_pre_processing`, `segmentation`, `tracking`, `time_series`, `spatial_graph`, `analysis_tools`, and `functional_metrics`. The first, `file_pre_processing`, is simply used to convert each movie into a series of 2D numpy arrays [23], one for each frame of the movie converted to grayscale. The code is designed such that the only new step required to support a previously unsupported movie file format is a function to convert each frame into a 2D numpy array. The remainder of the code can loosely be grouped as `segmentation` which is necessary for extracting spatial data, `tracking` and `time_series` which are necessary for extracting temporal data, and `spatial_graph`, `analysis_tools`, and `functional_metrics` which are necessary for synthesizing and interpreting the spatial and temporal data.

**Segmentation.** Within the `segmentation` script, the function `segmentation_all (folder_name, gaussian_filter_size)` will segment all z-discs and all sarcomeres from each frame. The first parameter, "`folder_name`" specifies the folder that the data is in. The second parameter, "`gaussian_filter_size`" specifies the standard deviation for the Gaussian kernel applied to the image during z-disc segmentation [30]. The default value for `gaussian_filter_size` = 1 and that is the value for all data shown in this paper. In our experience, a value of 2 may be more appropriate for movies of hiPSC-CMs that have high levels of irregular fluctuating background signal, and thus it is left as a potential tunable parameter with a default value of 1. The key steps of segmentation are illustrated in Fig 3. z-disc segmentation is conducted by computing the Laplacian of the image [27], applying a Gaussian filter to the Laplacian [30], and then using the scikit-image "`measure.`

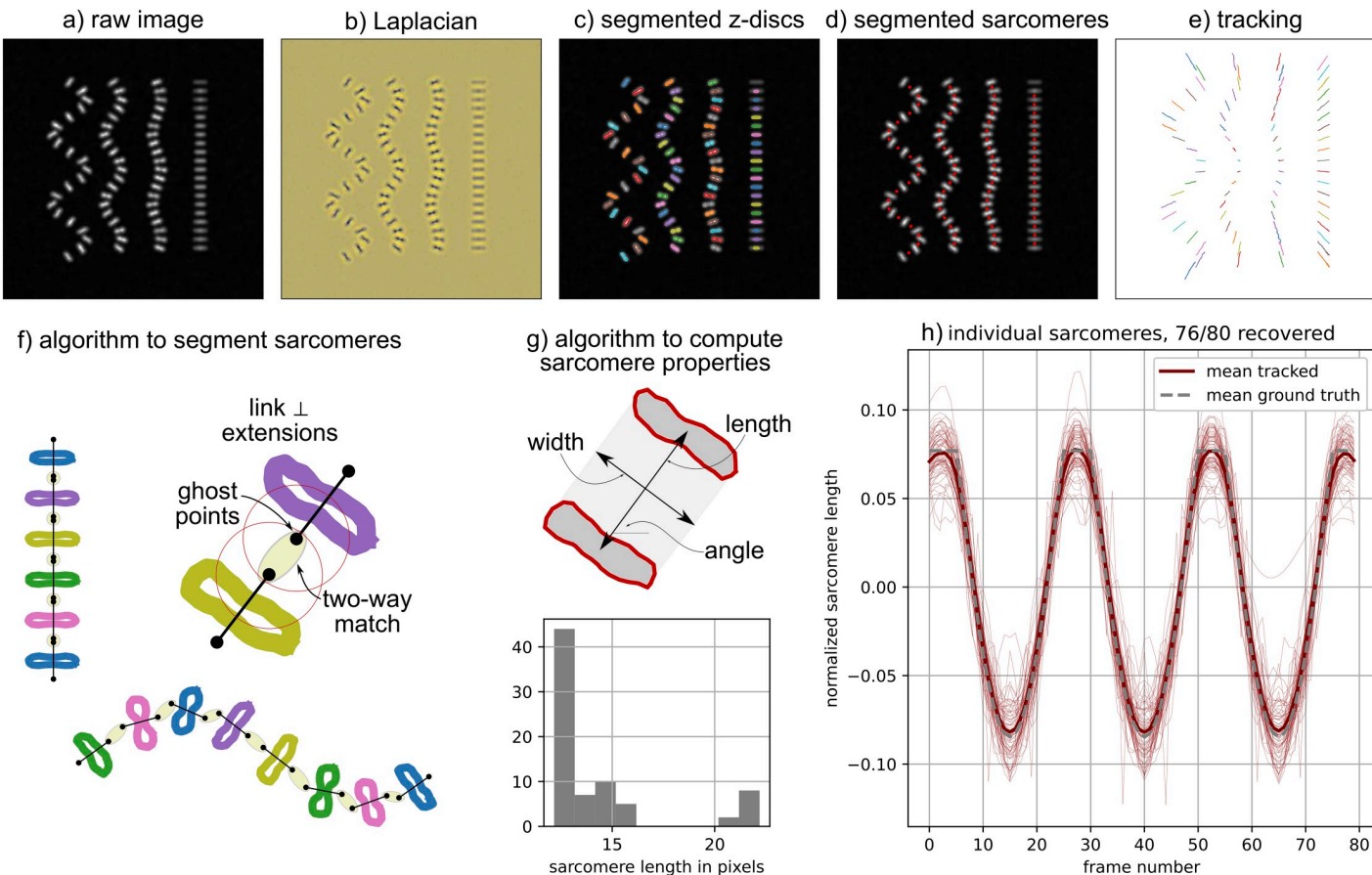

**Fig 3.** Segmenting and tracking the z-discs and sarcomeres: (a) An example of a raw synthetic two-dimensional (2D) input image; (b) The Laplacian of the input image is used to detect the high gradients present at the edge of every z-disc [27]; (c) z-discs are identified as closed contours where the value of the Laplacian exceeds the threshold computed with Otsu's method [28]; (d) Sarcomeres are procedurally identified from the segmented z-discs; (e) Sarcomeres and z-discs are tracked independently between frames with the Python trackpy package [29]; (f) The algorithm to segment sarcomeres is based on linking the approximately parallel z-discs to their closest neighbors in the direction perpendicular to the z-disc; (g) Sarcomere properties are computed from the pair of associated z-discs; (h) Tracking each sarcomere leads to multiple spatially resolved time series.

find_contours()" function [31] to identify contours with a level matching a scalar threshold computed via Otsu's method [28]. We note briefly that a global threshold determined with Otsu's method is sufficient (i.e., an adaptive threshold is not required) because thresholding is performed on the Laplacian rather than the original image. As illustrated in Fig 3C, each contour represents a segmented z-disc and the properties of each z-disc are computed algorithmically from the contours. Of note, contour position is computed as the mean position of each pixel in the contour, contour length is computed as the maximum possible distance between two pixels in the contour, and contour endpoints are the locations of the pixels that correspond to the maximum distance.

Once z-discs are segmented, sarcomeres are procedurally identified from the segmented z-discs with a custom algorithm schematically illustrated in Fig 3F. In brief, the steps of the algorithm are as follows:

1. Compute the distance between the center of each z-discs and the center of its nearest neighbor.

2. Define `median_neigh` as the median distance between a z-disc and its nearest neighbor.

3. For each z-disc, define a line that goes through the center of the z-disc and is perpendicular to the axis defined by the contour endpoints. Then, define the two points on the line that are `median_neigh`/2 distance from the z-disc center. These points are referred to as "`ghost_points`." This is illustrated in Fig 3F.

4. Find the nearest neighbor for each point in `ghost_points`.

5. If there is a two-way nearest neighbor match between a pair in `ghost_points`, the pair of z-discs will correspond to a segmented sarcomere.

As illustrated in Fig 3F, this algorithm links approximately parallel z-discs. In order to flexibly accommodate potential image artifacts, the criteria for "approximately parallel" is not strictly defined. However, in most cases, there will not be a persistent two way nearest neighbor match unless z-discs bound a convex quadrilateral, ideally a trapezoid. If necessary, additional match rejection criteria could be integrated into the algorithm. Once a sarcomere is identified, its properties are computed from the properties of the corresponding paired z-discs. This is illustrated in Fig 3G. Of note, sarcomere width is computed as the mean length of the two z-discs, sarcomere length is computed as the distance between the two z-disc centers, and sarcomere angle is computed as the angle of the vector connecting the two z-disc centers.

For the examples shown in this paper, the segmentation step takes order of 10s of seconds to a few minutes to run on a single laptop. We also note that users working with still images rather than movies can still segment z-discs and sarcomeres from a single frame, and can still create a spatial graph with their data and perform several aspects of the spatial analysis described in this paper. For generality, all length data is reported in units of pixels.

**Tracking and processing time series data.** The `tracking` component of Sarc-Graph is for independently tracking all z-discs and all sarcomeres. Tracking the sarcomeres is necessary for reporting the relative length change from frame to frame. Without tracking, it is only possible to report population average relative length change which is substantially less information rich because the sarcomeres do not necessarily contract in perfect synchrony, and are not necessarily of identical lengths. We track the z-discs and sarcomeres independently because we use this information to construct a spatial graph, described later in the text. Tracking is performed by calling the "`run_all_tracking(folder_name,tp_depth)`" function. As stated previously, "`folder_name`" specifies the folder that the data is in. The second parameter, "`tp_depth`" specifies the farthest distance in pixels that a tracked entity can travel between frames [29]. The default value for `tp_depth` = 4 and that is the value for all data shown in this paper. In some cases, particularly for movies with lower spatial resolution, it is necessary to change to `tp_depth` = 3. Particle tracking is performed with the Python package trackpy which is based on the Crocker–Grier algorithm [29, 32]. We note that we do not use trackpy for feature detection, instead we convert each segmented z-disc and sarcomere to a pandas DataFrame for compatibility with the trackpy framework [33]. The outcome of running `run_all_tracking()` is a unique ID for each particle that is tracked for over 10% of the movie. With this information, it is possible to locate each tracked particle in space across multiple frames, as illustrated in Fig 3E where the position of each sarcomere across all frames is plotted.

The `time_series` component of our computational framework processes the results from `tracking` so that there is time series data describing the length, normalized length $(L - \bar{L})/\bar{L}$, width, angle, and position for each tracked sarcomere. This is accomplished by running the function "`timeseries_all(folder_name, keep_thresh)`" where `keep_thresh` represents the fraction of movie frames that a tracked sarcomere must be

present in to get processed. We recommend `keep_thresh` = 0.75 for quantitative analysis of time series data and `keep_thresh`>1/$k$ for visualization, where $k$ is the approximate number of times the cell contracts during one movie. To process the results from `tracking`, Gaussian process regression is used to interpolate data across frames which temporarily lose signal [34]. Gaussian process regression is implemented through python scikit-learn with a radial basis function (RBF) and white noise kernel [35]. In addition to interpolating between lost frames, Gaussian process regression has the added benefit of adaptively smoothing the data without a need for additional dataset-specific input parameters. The main result of running the `timeseries_all` function is illustrated in Fig 3H where normalized sarcomere length is plotted with respect to frame number. Even with this synthetic data example, the individual recovered time series deviate from the ground truth. This is for two main reasons. First, for the ground truth geometry that we specified, some sarcomeres are angled out of the image z-plane and are thus not perfectly captured by a 2D image. Second, the limited resolution of the images introduces artifacts where lengths less than 1 pixel cannot be perfectly captured. However, the mean of all time series curves illustrated in Fig 3H is a near perfect fit to the ground truth.

For the examples shown in this paper, the `tracking` step takes order of 10s of seconds to a minute to run on a single laptop. The `time_series` step takes on the order of 10s of seconds to a few minutes to run. The `time_series` step will output several user friendly text files describing sarcomere properties with respect to time where each row corresponds to a tracked sarcomere and each column corresponds to a movie frame. The normalized length data can be automatically plotted with the function `plot_normalized_tracked_ti-meseries()` from the `analysis_tools` file. For generality, all time data is reported in units of frames.

**Computing average deformation ($\mathbf{F}_{avg}$).** Developing methods to describe data rich images and movies of cells with a tractable set of output parameters is an active area of research [11, 36]. In this paper specifically, we draw inspiration from recent work on summarizing data from traction force microscopy experiments [20] and agent based model simulations [37, 38] and propose a method to compute the mean deformation gradient of each domain with respect to time. First, we define the standard continuum mechanics deformation gradient $\mathbf{F}$ as follows:

$$\mathbf{F}\, d\mathbf{X} = d\mathbf{x} \tag{1}$$

where $d\mathbf{X}$ is a vector in the initial configuration, $d\mathbf{x}$ is the vector in the deformed configuration, and $\mathbf{F}$ is the mapping between them [39]. In the context of analyzing movies of contracting hiPSC-CMs, we first define a set of $n$ vectors $\mathbf{v}$ that connect each potential pair of tracked fiducial markers, as illustrated in Fig 4A. The direction and magnitude of each vector $\mathbf{v}$ will change as the fiducial markers move between movie frames. With this definition, we can set up the over-determined system of equations:

$$\mathbf{F}_{avg}\,\boldsymbol{\Lambda}_0 = \boldsymbol{\Lambda} \qquad \text{where} \qquad \boldsymbol{\Lambda}_0 = [\mathbf{v}_{01}, \mathbf{v}_{02}, \ldots, \mathbf{v}_{0n}] \qquad \text{and} \qquad \boldsymbol{\Lambda} = [\mathbf{v}_1, \mathbf{v}_2, \ldots, \mathbf{v}_n] \tag{2}$$

where $\mathbf{F}_{avg}$ is a $2 \times 2$ matrix, $\boldsymbol{\Lambda}_0$ is a $2 \times n$ matrix of vectors in the initial (reference) configuration movie frame, and $\boldsymbol{\Lambda}$ is a $2 \times n$ matrix of vectors in the current (deformed) configuration movie frame. Note that when the initial frame and the current frame are identical, $\boldsymbol{\Lambda}_0 = \boldsymbol{\Lambda}$ and $\mathbf{F}_{avg} = \mathbf{I}$. The initial configuration can be defined as either the first frame of the movie, or a selected movie frame where the cell is known to be in a relaxed state. Then, we can use the

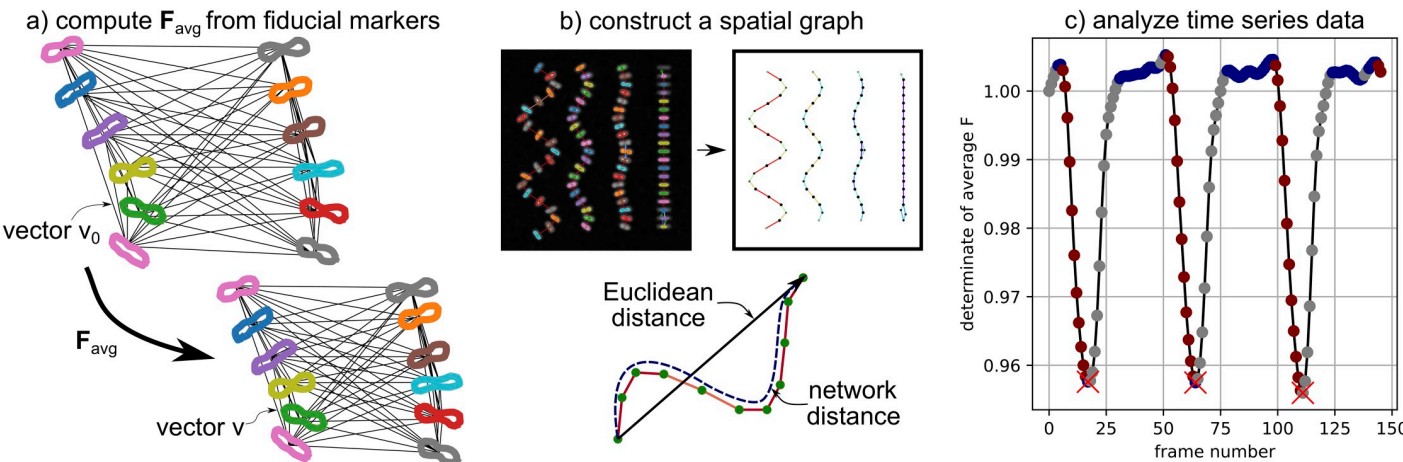

**Fig 4.** Data analysis from segmentation and tracking: (a) Tracked elements are treated as fiducial markers and used to construct an average deformation gradient **F** (see Eqs 1–4); (b) z-discs and sarcomeres are tracked independently and used to construct a spatial graph (z-discs are "nodes" and sarcomeres are "edges", edge color corresponds to sarcomere orientation, node color corresponds to correlation in sarcomere orientation) to measure correlations based on network distance; (c) Automated analysis of time series data is used to extract constants such as mean contraction time and number of peaks.

normal equation to solve for the best fit average deformation gradient as:

$$\mathbf{F}_{\mathrm{avg}} = \mathbf{\Lambda}\mathbf{\Lambda}_0^{\mathrm{T}} \left[ \mathbf{\Lambda}_0\mathbf{\Lambda}_0^{\mathrm{T}} \right]^{-1} . \tag{3}$$

Once computed, the components of $\mathbf{F}_{\mathrm{avg}}$ can be plotted directly, or $\mathbf{F}_{\mathrm{avg}}$ can be manipulated to provide information that is more convenient to compare and interpret. For example, we can write

$$\mathbf{F}_{\mathrm{avg}} = \mathbf{R}_{\mathrm{avg}}\mathbf{U}_{\mathrm{avg}} \tag{4}$$

where $\mathbf{R}_{\mathrm{avg}}$ is a rotation tensor and $\mathbf{U}_{\mathrm{avg}}$ is a symmetric positive definite tensor that represents stretch. In the numerical setting, $\mathbf{R}_{\mathrm{avg}}$ and $\mathbf{U}_{\mathrm{avg}}$ are computed with the scipy polar decomposition function [30]. We can also compute the eigenvalues and eigenvectors of $\mathbf{U}_{\mathrm{avg}}$, $\lambda_1^{\mathrm{avg}}$, $\lambda_2^{\mathrm{avg}}$, $\mathbf{u}_1^{\mathrm{avg}}$, and $\mathbf{u}_2^{\mathrm{avg}}$ respectively. In the numerical setting, this is done with the numpy linalg package, and we always define $\lambda_1^{\mathrm{avg}} < \lambda_2^{\mathrm{avg}}$ [23]. From a mechanical data interpretation perspective, the average stretch values $\lambda_1^{\mathrm{avg}}$ and $\lambda_2^{\mathrm{avg}}$ and the associated eigenvectors are particularly convenient because they contain information about both the magnitude and direction of contraction. This is illustrated in Fig 1B and 1C.

Because sarcomere data is already processed through the `time_series` component of our framework, we use the sarcomeres as our fiducial markers to define $\mathbf{\Lambda}$ and $\mathbf{\Lambda}_0$. We note that this approach would be just as effective with the z-discs chosen as fiducial markers. In addition, if 3D data was available, this approach would still work with $\mathbf{F}_{\mathrm{avg}}$ as a $3 \times 3$ matrix and $\mathbf{\Lambda}$ and $\mathbf{\Lambda}_0$ as $3 \times n$ matrices. Within the `analysis_tools` file, the function `compute_F_whole_movie()` will compute $\mathbf{F}_{\mathrm{avg}}$ for each movie frame. The function `visualize_F_full_movie()` will create a visualization of $\lambda_1^{\mathrm{avg}}$ and $\lambda_2^{\mathrm{avg}}$ as a function of the movie frame and schematically illustrate $\mathbf{F}_{\mathrm{avg}}$ overlaid on the original image data, as shown in Fig 1. The computational cost of computing and processing $\mathbf{F}_{\mathrm{avg}}$ for all frames is on the order of seconds to 10s of seconds on a single laptop. The computational cost of data visualization is on the order of 10s of seconds to minutes.

**Comparing average deformation ($\mathbf{F}_{avg}$) to average sarcomere shortening ($\tilde{s}$, $s_{avg}$) and Orientational Order Parameter (OOP).** In addition to defining tensor $\mathbf{F}_{avg}$ as a dynamic metric that evolves over the course of the entire beating hiPSC-CM movie, it is possible to formulate scalar metrics from $\mathbf{F}_{avg}$ that are more straightforward to directly compare to well established metrics for describing hiPSC-CMs [7]. Here we focus on defining scalar metrics from $\mathbf{F}_{avg}$ that are directly comparable to average sarcomere shortening ($\tilde{s}$, $s_{avg}$) [10], and the Orientational Order Parameter (OOP) [11, 14, 40].

We compute average sarcomere shortening in two different ways. In both cases, we work with the normalized sarcomere length obtained from `time_series` $y = (L - \bar{L})/\bar{L}$ where $L$ is sarcomere length in pixels. Given normalized sarcomere length $y$ for each movie frame, we define shortening as:

$$s = \frac{y_{max} - y_{min}}{y_{max} + 1} \tag{5}$$

where $y_{max}$ is the maximum normalized length and $y_{min}$ is the minimum normalized length. We note briefly that this is equivalent to defining $s = (L_{max} - L_{min})/L_{max}$. Then, we can define $\tilde{s}$ as the median value of $s$ for all sarcomeres tracked in the movie. We choose to use median rather than mean in this case because the mean is more susceptible to outliers and thus $\tilde{s}$ is a better reflection of the ground truth sarcomere deformation. In addition, we can compute the mean normalized length time series $y_{avg}$ and then compute $s_{avg}$ based on the single mean time series. If all sarcomeres are contracting identically, $\tilde{s}$ and $s_{avg}$ will be identical. However, if this is not the case, the two values will likely differ and $\tilde{s}$ will reflect average individual sarcomere behavior while $s_{avg}$ will reflect both individual sarcomeres and the (lack of) synchrony between them. Functions to compute $\tilde{s}$ and $s_{avg}$ are given in the `functional_metrics` component of Sarc-Graph.

In addition to computing mean sarcomere shortening, we specify the method for computing OOP from hiPSC-CM movies that we implement in the `functional_metrics` component of Sarc-Graph. When hiPSC-CMs are fully relaxed, individual sarcomere chains are no longer under systolic tension and can thus become wavy and lose local alignment. Therefore, the first step to computing OOP is to define a frame of the movie where the lowest fraction of sarcomere chains will be in their relaxed and potentially wavy configuration. To select this frame, we compute $\det(\mathbf{F}_{avg})$ for every movie frame with the first frame of the movie used as the reference configuration. Then, we identify the frame where $\det(\mathbf{F}_{avg})$ is the greatest (i.e. the sarcomeres are most relaxed) and select that frame as the reference frame and recompute $\mathbf{F}_{avg}$. With the appropriately selected reference frame, we re-compute $\det(\mathbf{F}_{avg})$ and define the most contracted frame as the frame with the lowest value of $\det(\mathbf{F}_{avg})$.

We compute OOP for the static image corresponding to the most contracted frame of the movie. To do this, we define structural tensor $\mathbb{T}$ following the literature:

$$\mathbb{T} = \left\langle 2 \begin{bmatrix} r_x^i r_x^i & r_x^i r_y^i \\ r_y^i r_x^i & r_y^i r_y^i \end{bmatrix} - \begin{bmatrix} 1 & 0 \\ 0 & 1 \end{bmatrix} \right\rangle \tag{6}$$

where $\mathbf{r}^i = [r_x^i, r_y^i]$ is the unit vector describing the orientation of the $i^{th}$ sarcomere [14]. The structural tensor contains information about the amount and direction of sarcomere alignment. Given $\mathbb{T}$, with eigenvalues $a_{max}$ and $a_{min}$ and corresponding unit eigenvectors $\mathbf{v}_{max}$ and $\mathbf{v}_{min}$, OOP = $a_{max}$. When OOP = 0.0, sarcomeres are oriented randomly. When OOP = 1.0, sarcomeres are perfectly aligned. Corresponding eigenvector $\mathbf{v}_{max}$ corresponds to the dominant direction of sarcomere orientation.

In order to best relate $\mathbf{F}_{avg}$ to OOP, we define two scalar metrics $C_{iso}$ and $C_{\parallel}$ that describe average radial contraction, and contraction in the direction of dominant sarcomere orientation $\mathbf{v}_{max}$ respectively. Unlike individual sarcomere contraction or the average of individual sarcomere contraction, $C_{iso}$ and $C_{\parallel}$ represent deformation throughout the whole domain and not necessarily along the sarcomere axis. Metric $C_{iso}$ is defined as

$$C_{iso} = 1.0 - \sqrt[\text{dim}]{\det(\mathbf{F}_{avg})} \tag{7}$$

where $\mathbf{F}_{avg}$ is computed in the most contracted frame and dim = 2 for the results presented in this paper (for extension to 3$D$, dim = 3). Practically, $C_{iso}$ is the line shortening during contraction of an equivalent system where contraction is not directionally dependent. When $C_{iso}$ = 0.0, no shortening occurs. As an example, a value $C_{iso}$ = 0.10 corresponds to an equivalent isotropic line shortening of 10% in all directions. Essentially, higher $C_{iso}$ indicates higher levels of hiPSC-CM contraction.

To quantify shortening in the dominant direction specified by $\mathbf{v}_{max}$ computed from $\mathbb{T}$, we consider the relation $\mathbf{F}_{avg}\mathbf{v}_0 = \mathbf{v}_{max}$ where $\mathbf{v}_0$ is the corresponding vector defined in the reference frame. We can manipulate this equation to compute $\mathbf{v}_0 = \mathbf{F}_{avg}^{-1}\mathbf{v}_{max}$ and then define $C_{\parallel}$ as

$$C_{\parallel} = \frac{|\mathbf{v}_0| - |\mathbf{v}_{max}|}{|\mathbf{v}_0|} \ . \tag{8}$$

With this definition, $C_{\parallel}$ is the fractional shortening of a line in the direction of $\mathbf{v}_{max}$ where direction is defined in the contracted configuration. Conveniently, lower $\tilde{s}$ and $s_{avg}$, $C_{iso}$, and $C_{\parallel}$ all correspond to lower levels of contraction, and higher $\tilde{s}$ and $s_{avg}$, $C_{iso}$, and $C_{\parallel}$ all correspond to higher levels of contraction.

Functions to compute and visualize $\tilde{s}$, $s_{avg}$, OOP, $C_{iso}$, and $C_{\parallel}$ are all defined in the `functional_metrics` script. The function "`compute_metrics`" will compute $\tilde{s}$, $s_{avg}$, OOP, $C_{iso}$, and $C_{\parallel}$ and create a visualization of OOP and $\mathbf{F}_{avg}$. The function "`visualize_lambda_as_functional_metric`" will create a movie of $\lambda_1^{avg}$ and $\lambda_2^{avg}$ changing over the course of the movie. Examples of this are provided as Supplementary Information. The computational cost of computing $\mathbf{F}_{avg}$ (including computing $C_{iso}$, and $C_{\parallel}$) and $\tilde{s}$, $s_{avg}$, and OOP for all frames is on the order of seconds to 10s of seconds on a single laptop.

**Preparing spatial graphs.** In addition to analyzing average cell and sarcomere behavior (ex: average z-disc and sarcomere morphological properties, $\mathbf{F}_{avg}$), we are interested in analyzing the spatiotemporal patterns that arise on the sub-cellular level. We implemented the `spatial_graph` component of Sarc-Graph to conveniently synthesize the outputs of the `segmentation`, `tracking`, and `time_series` steps. Specifically, running the function `create_spatial_graph()` will create a spatial graph where z-discs are represented as nodes and sarcomeres are represented as edges. With a spatial graph structure, it is then possible to better quantify spatiotemporal patterns in hiPSC-CM behavior. For example, access to a spatial graph can help delineate sarcomere synchrony that depends on Euclidean distance, and sarcomere synchrony that depends on network distance. In addition, access to a spatial graph can help us better quantify both global and local sarcomere alignment. This functionality is illustrated in Fig 4B.

In the `run_all_tracking()` step, z-discs and sarcomeres are tracked independently. Every z-disc that is at least partially tracked (defined as present in over 10% of the movie frames) will have a unique local (frame-specific) ID and a unique global ID. Each node of the spatial graph corresponds to a unique global ID, and the local IDs specific to each frame are linked to the corresponding global ID node. Each tracked sarcomere is associated with two local z-disc IDs in each frame. To add sarcomeres to the spatial graph, the global ID of the

associated z-disc is first determined. Then, either a new edge is created linking the two global IDs, or, if the edge already exists, the weight of the edge is increased. The spatial graph is created with the NetworkX python package, and all operations, such as computing the shortest path between two nodes, rely on built-in NetworkX functionality [41]. The function `create_spatial_graph()` runs in seconds on a single laptop. Running `create_spatial_graph()` will also produce a schematic drawing of the spatial graph.

**Notable analysis and visualization tools.** In addition to the functions to compute and visualize $\mathbf{F}_{avg}$ and other functional metrics (`compute_F_whole_movie()`, `visualize_F_full_movie()`, `compute_metrics()`, `visualize_lambda_as_functional_metric()`) and plot normalized sarcomere length with respect to time (`plot_normalized_tracked_timeseries()`), the `analysis_tools` component of Sarc-Graph contains multiple functions to further visualize and analyze the data acquired from the hiPSC-CM movies. Example outputs from these functions are included on the Sarc-Graph GitHub page. Additional key analysis and visualization functions are as follows:

- `visualize_segmentation(folder_name, gaussian_filter_size, frame_num)`: Visualization of the segmented z-discs and sarcomeres overlaid on the original image. The parameter `gaussian_filter_size` should match the value chosen in `segmentation`. The default for `gaussian_filter_size = 1` and the default for `frame_num = 0`.

- `visualize_contract_anim_movie(folder_name, re_run_timeseries, use_re_run_timeseries, keep_thresh = 0.75)`: Visualization of the results of tracking overlaid on the original movie of the contracting hiPSC-CMs. If `re_run_timeseries = True` and `use_re_run_timeseries = True` this function will re-run the `timeseries_all()` function with the new specified `keep_thresh`. The saved tracking data will be designated for visualization purposes only.

- `cluster_timeseries_plot_dendrogram(folder_name, compute_dist_DTW, compute_dist_euclidean)`: Visualize hierarchical clustering of all normalized sarcomere length time series data and plot a dendrogram that illustrates the clustering [30]. If `compute_dist_DTW = True` the similarity between each sarcomere time series will be measured with a Dynamic Time Warping algorithm [42]. Because this can be time consuming (order of minutes to 10s of minutes), we also provide an option to set `compute_dist_euclidean = True` and instead base clustering on the Euclidean distance between time series.

- `plot_normalized_tracked_timeseries(folder_name)`: This function will create a plot of the normalized length of the tracked sarcomeres with respect to frame number.

- `plot_untracked_absolute_timeseries(folder_name)`: This function will create a plot of the absolute sarcomere lengths in pixels with respect to frame number for all segmented sarcomeres. The number of segmented sarcomeres will exceed the number of tracked sarcomeres.

- `compute_timeseries_individual_parameters(folder_name)`: This function will compute and save scalar values describing the normalized length time series data for the tracked sarcomeres (contraction time, relaxation time, flat time, period, offset, etc.) The results of this analysis will be saved in a spreadsheet titled "`timeseries_parameters_info.xlsx`."

- `analyze_J_full_movie(folder_name)`: This function will compute and save scalar values describing the $\mathbf{F}_{avg}$ time series data. Specifically, it will compute time constants describing the plot of the scalar Jacobian($J = \det \mathbf{F}_{avg}$) with respect to frame number. This is illustrated in Fig 3C.

- `compare_tracked_untracked(folder_name)`: This function will compare the tracked and untracked populations through random sampling of the untracked population. These plots are important for understanding if the tracked data is significantly biased compared to the segmented data.

- `preliminary_spatial_temporal_correlation_info(folder_name)`: This function will perform a preliminary analysis of spatial/temporal correlation within the hiPSC-CMs. The main outcome is a plot of normalized cross-correlation score between normalized sarcomere length time series data with respect to both Euclidean and network distances.

## Results and discussion

### Synthetic data examples

Our first investigation into the performance of Sarc-Graph on synthetic data is illustrated in Fig 5. In this investigation, we define a sinusoidal chain of 20 sarcomeres that deform homogeneously following the time series illustrated in Fig 2A. Under baseline conditions (modest amplitude, little angling out of plane, and low to moderate noise), Sarc-Graph is able to successfully track all 20 out of 20 sarcomeres. This is illustrated in Fig 5A. Then, we adversely alter these baseline conditions to demonstrate the limitations of our approach. In the first study, illustrated in Fig 5B, we alter the underlying 3D geometry of the sarcomere chain. As anticipated, performance degrades when we increase the amplitude of the sine wave, and when we

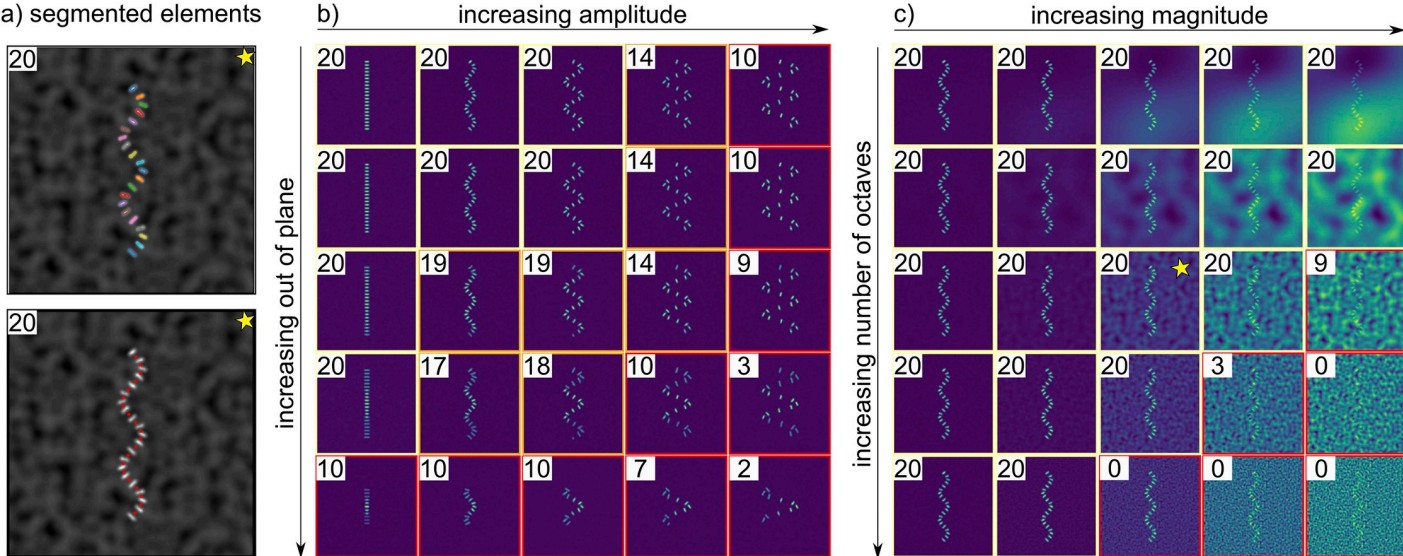

**Fig 5. For each image, the number of sarcomeres successfully segmented and tracked (out of 20 possible) is reported.** Running the algorithm on synthetic data shows that the code is robust to geometry and noise up to a limit: (a) Illustration of successful z-disc and sarcomere segmentation and tracking for a curved geometry in the presence of noise; (b) Performance degrades when the baseline geometry moves partially out of plane (here slanted in the z-direction), and when the sarcomere chain is too distorted (here high amplitude to period ratio for sinusoidal geometry); (c) Performance degrades in the presence of Perlin noise, in particular noise that is similar in size and brightness to the z-discs themselves.

increase the out of plane tilt of the sarcomere chain. That being said, we are able to segment and track at least half of the 20 sarcomeres in all but four of the examples shown. In the second study, illustrated in Fig 5C, we adversely alter baseline conditions by adding noise to the rendered image. Specifically, we add Perlin noise at varying levels of magnitude and with a varying number of octaves [22]. From these results, it is clear that our computational framework is quite robust to all tested types of low magnitude noise, and Perlin noise with a low number of octaves (low frequency). However, performance degrades in the presence of Perlin noise that has high gradients and is similar in size and brightness to the simulated z-discs themselves. We note that if a specific experimental dataset has a characteristic noise pattern that can be removed with a known operation, implementing additional steps in the `segmentation` component of our computational framework to address it would be straightforward. In some cases, simply increasing the parameter `gaussian_filter_size` will resolve the issue. In the experimental data that we have tested the computational framework on thus far, our current segmentation process is effective. To complement the synthetic data results shown in Fig 5, S3 Text shows the effect of decreasing image resolution on Sarc-Graph performance.

Our second investigation into the performance of Sarc-Graph on synthetic data is illustrated in Fig 6. In these examples, our aim is to design synthetic data that contains many of challenges associated with analyzing real hiPSC-CMs: out of plane deformation, sarcomere chain curvature, sarcomere chain overlap, inhomogeneous deformation, variable z-disc size, and irregular contraction. From S1 to S5 the examples are loosely organized from "least" to "most" challenging for the computational framework to capture. The corresponding movies are included in our GitHub repository. Briefly, example S1 shows four sarcomere chains of variable curvature (decreasing from left to right) and variable out of plane deformation (increasing from left to right) deforming homogeneously. Our framework is able to recover 77 out of 80 sarcomeres, match the ground truth mean time series data, and match the ground truth on $\lambda_1^{\mathrm{avg}}$ and $\lambda_2^{\mathrm{avg}}$. This is consistent with the results shown in Fig 5. Example S2 shows two families of sarcomere chains, an external ellipse and four internal chains that overlap in the center. Our framework is able to recover 90 out of 99 sarcomeres (error occurs primarily near the overlap region), match the ground truth mean time series data, and match the ground truth on $\lambda_1^{\mathrm{avg}}$ and $\lambda_2^{\mathrm{avg}}$.

Example S3 shows three closely positioned elliptical sarcomere chains that deform inhomogeneously. Our framework is able to recover 34 out of 40 sarcomeres, however, because there is bias in which sarcomeres are recovered (recovery is worse towards the bottom of the image frame), both the mean time series data and the $\lambda_1^{\mathrm{avg}}$ and $\lambda_2^{\mathrm{avg}}$ data are unable to perfectly capture the ground truth. We note that the results for $\lambda_1^{\mathrm{avg}}$ and $\lambda_2^{\mathrm{avg}}$ are a better reflection of the ground truth than the mean time series data. We also note that the amount of sarcomere contraction simulated in S3 exceeds what is typically observed in the experimental setting (over 30% contraction).

Example S4 shows two families of sarcomere chains: an external ellipse, and multiple partially overlapping and curved internal chains in different z-planes. In Example S4, the sarcomeres deform inhomogeneously with a brief abrupt jump in sarcomere deformation around frame 70. Our framework is able to recover 75 out of 90 sarcomeres, and is able to nearly recover the mean time series data, and the $\lambda_1^{\mathrm{avg}}$ and $\lambda_2^{\mathrm{avg}}$ data. We note that for both approaches, the abrupt jump in sarcomere behavior seen in the ground truth is smoothed out in the automated time series processing because during the abrupt jump tracking temporarily fails for the entire outer ring of sarcomeres. There are two main implications of this result. First, that large abrupt motion may cause Sarc-Graph to fail. Second, that Sarc-Graph is able to track sarcomeres on either end of a time period where signal is temporarily lost, and interpolate data

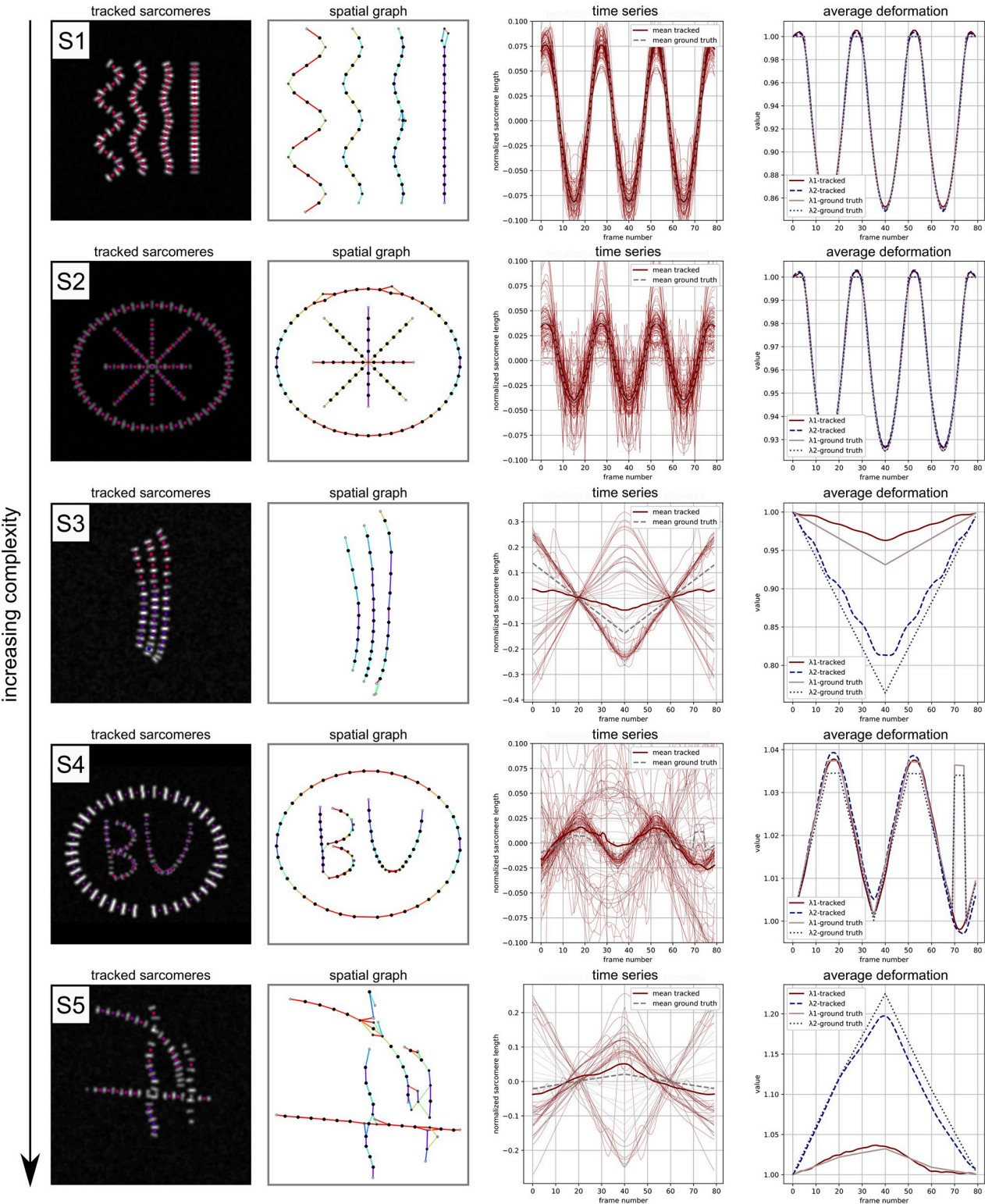

**Fig 6.** Example performance on synthetic data of increasing complexity with a known ground truth: Segmentation and tracking (marker color corresponds to sarcomere contraction in the illustrated frame); Constructing a spatial graph (line color corresponds to sarcomere orientation); Recovering individual time series data (both Sarc-Graph output and ground truth shown); Recovering average deformation behavior (both Sarc-Graph output and ground truth shown).

**Table 2. Comparison of Sarc-Graph computed $\tilde{s}$, $s_{avg}$, OOP, $C_{iso}$, and $C_{\parallel}$ with the ground truth values of these parameters for each synthetic data example.**

|    | $\tilde{s}$ | $\tilde{s}^{GT}$ | $s_{avg}$ | $s_{avg}^{GT}$ | OOP | OOP$^{GT}$ | $C_{iso}$ | $C_{iso}^{GT}$ | $C_{\parallel}$ | $C_{\parallel}^{GT}$ |
|----|------|------|------|------|------|------|------|------|------|------|
| S1 | 0.15 | 0.15 | 0.15 | 0.15 | 0.62 | 0.63 | 0.15 | 0.15 | 0.15 | 0.15 |
| S2 | 0.085 | 0.074 | 0.074 | 0.075 | 0.080 | 0.066 | 0.076 | 0.075 | 0.076 | 0.075 |
| S3 | 0.38 | 0.40 | 0.079 | 0.24 | 0.93 | 0.88 | 0.11 | 0.16 | 0.039 | 0.070 |
| S4 | 0.058 | 0.043 | 0.042 | 0.024 | 0.16 | 0.15 | 0.04 | 0.04 | 0.039 | 0.038 |
| S5 | 0.22 | 0.22 | 0.085 | 0.042 | 0.35 | 0.21 | 0.13 | 0.16 | 0.17 | 0.20 |

across the missing frames. Similar to example S3, the plot of $\lambda_1^{avg}$ and $\lambda_2^{avg}$ is a better reflection of the ground truth than the mean time series curve.

Example S5 shows multiple sarcomere chains that are all angled out of plane, appear to overlap at multiple points, and undergo inhomogeneous deformation. In this case, we are only able to recover time series data for 27 out of 70 sarcomeres. This results in mean time series data that is a poor reflection of the ground truth. However, the $\lambda_1^{avg}$ and $\lambda_2^{avg}$ data is a much better match to the ground truth. This is a positive sign towards the efficacy of our proposed metric even when the number of sarcomeres recovered is relatively low. Though only 27 out of 70 sarcomeres were fully tracked, segmented but only partially tracked sarcomeres appear on the spatial graph shown in Fig 6 example S5. Looking forward, additional criteria to further refine the spatial graph and further delineate "probable" and "improbable" links can readily be added to Sarc-Graph within the spatial_graph module.

In Table 2, we report median individual sarcomere shortening ($\tilde{s}$), mean time series sarcomere shortening ($s_{avg}$), Orientational Order Parameter (OOP), equivalent isotropic contraction ($C_{iso}$), and contraction in the dominant direction of sarcomere orientation ($C_{\parallel}$) for each synthetic data example. For all of the synthetic data cases, we are able to compute a ground truth for each of these parameters from the known ground truth contraction, displacement, and orientation of the sarcomere geometry. For examples S1 and S2, the parameters recovered by Sarc-Graph are a near perfect match to the ground truth. However, for all other samples minor discrepancies arise in $\tilde{s}$, OOP, $C_{iso}$, and $C_{\parallel}$. For example S4, $\tilde{s}$ is overestimated. For examples S3 and S5, OOP is overestimated, $C_{iso}$ is underestimated, and $C_{\parallel}$ is underestimated. Consistent with the results shown in Fig 6, we believe that this error is due to bias in the region where sarcomeres are recovered in S3, and failure of the algorithm for overlapping and severely out of plane sarcomere chains in S4 and S5. We note that for all examples but S1 and S2—where sarcomere contraction is synchronous—recovered $s_{avg}$ is not a good match to the ground truth. This is consistent with the normalized sarcomere length time series plot shown in Fig 6. However, the other metrics are able to perform acceptably in the presence of non-synchronous contraction. In general, these errors indicate that care should be taken when comparing results across beating hiPSC-CM movies that may have systemic differences. The accessibility of our code for generating additional synthetic data can aid in this preliminary check.

In all examples, most glaringly example S5, more sarcomeres are segmented and tracked for a small number of frames (and thus appear on the spatial graph) than are tracked for enough frames to reconstruct the full time series. The recovered time series ($\lambda_1^{avg}$ and $\lambda_2^{avg}$) and parameters ($C_{iso}$ and $C_{\parallel}$) associated with $\mathbf{F}_{avg}$ are typically a closer match to the synthetic data ground truth behavior than the mean curve of the recovered time series data. Example S5—with out of plane inhomogeneous deformation and overlapping sarcomere chains—is perhaps the best reflection of the challenges that we have seen in our experience thus far with the experimental data. We note that the extensible code used to generate the synthetic data is published on

GitHub so implementing and testing additional examples is straightforward. To complement these results, S4 Text shows the performance of Sarc-Graph on the synthetic data generated for the SarcTrack paper [10] and directly compares results from both Sarc-Graph and SarcTrack.

## Experimental data examples

Here, we show the performance of Sarc-Graph on the five experimental data examples listed in Table 1. In Figs 7–9, we illustrate the key Sarc-Graph outputs. In Table 3, we report $\tilde{s}$, $s_{avg}$, OOP, $C_{avg}$, and $C_{\parallel}$ for each example. In the Supplementary Information section, we include a movie for each sample that shows both the change in length of the successfully tracked sarcomeres, and how our computed metric $\mathbf{F}_{avg}$ changes over the course of the movie.

In Fig 7, we show the performance of Sarc-Graph on two similar examples of beating hiPSC-CM movies, examples E1 and E2. For example E1, we are able to segment $\approx 500$ sarcomeres per frame and successfully track 74 sarcomeres. The time series plot of absolute sarcomere length change with respect to frame number shows three distinct contraction events during the movie. Though the individual sarcomeres are clearly not perfectly in sync, there is enough of a unifying pattern for these three peaks to emerge. These three distinct contraction events are also reflected in the plot of $\lambda_1^{avg}$ and $\lambda_2^{avg}$. For example E2, we are able to segment $\approx 500$ sarcomeres per frame and successfully track 60 sarcomeres. The time series plots of absolute sarcomere length change and $\lambda_1^{avg}$ and $\lambda_2^{avg}$ with respect to frame number also show three distinct peaks. We also show the spatial graph representation of each example and a plot of normalized cross-correlation between pairs of individual sarcomere time series curves with respect to network distance.

Qualitatively, examples E1 and E2 are different in that the sarcomeres in example E1 appear vertically(top/bottom of the page) aligned throughout while the sarcomeres in example E2 appear tangentially aligned around the edge of the cell and unaligned in the center. This qualitative observation is consistent with the values of OOP reported in Table 3. In addition, we can use our computed metric $\mathbf{F}_{avg}$ to go beyond morphology alone and quantify the function of the observed hiPSC-CMs. By looking at the time series plots of $\lambda_1^{avg}$ and $\lambda_2^{avg}$ with respect to frame number, we can see that example E1 is deforming quite anisotropically $\lambda_1^{avg} < \lambda_2^{avg}$ while the deformation of example E2 is much closer to isotropic where $\lambda_1^{avg} \approx \lambda_2^{avg}$. In example E1, $C_{\parallel}$ is meaningfully greater than $C_{iso}$ while in example E2 the two parameters are much closer in value. Essentially, example E1 is experiencing more substantial oriented contraction than example E2. We also note that for these two examples, where $\tilde{s}$ is similar but OOP is different, metrics derived from $\mathbf{F}_{avg}$ are able to capture functional differences in contraction.

In Fig 8, we show the performance of Sarc-Graph on two contrasting examples of beating hiPSC-CM movies, examples E3 and E4. Example E3 is of a single hiPSC-CM adhered to a patterned soft hydrogel substrate with highly aligned sarcomeres and large deformation while Example E4 is of hiPSC-CMs in a monolayer culture without a clear direction of sarcomere alignment. With Sarc-Graph, we are able to segment approximately 100, and track 44 sarcomeres from E3. From the plots of $\lambda_1^{avg}$ and $\lambda_2^{avg}$, and scalar metrics OOP, $C_{iso}$, and $C_{\parallel}$, we observe high sarcomere alignment and corresponding highly aligned contraction. Notably, the cell contracts substantially in the direction perpendicular to fiber alignment, thus exhibiting auxetic deformation during contraction. For example E4 we are able to segment approximately 700, and track 228 sarcomeres. This is notable because the conditions in E4 are far from ideal for image analysis. In both cases, five distinct contraction events are observed over the course of the movie and are visible from both the average normalized sarcomere length time series and the $\lambda_1^{avg}$ and $\lambda_2^{avg}$ time series.

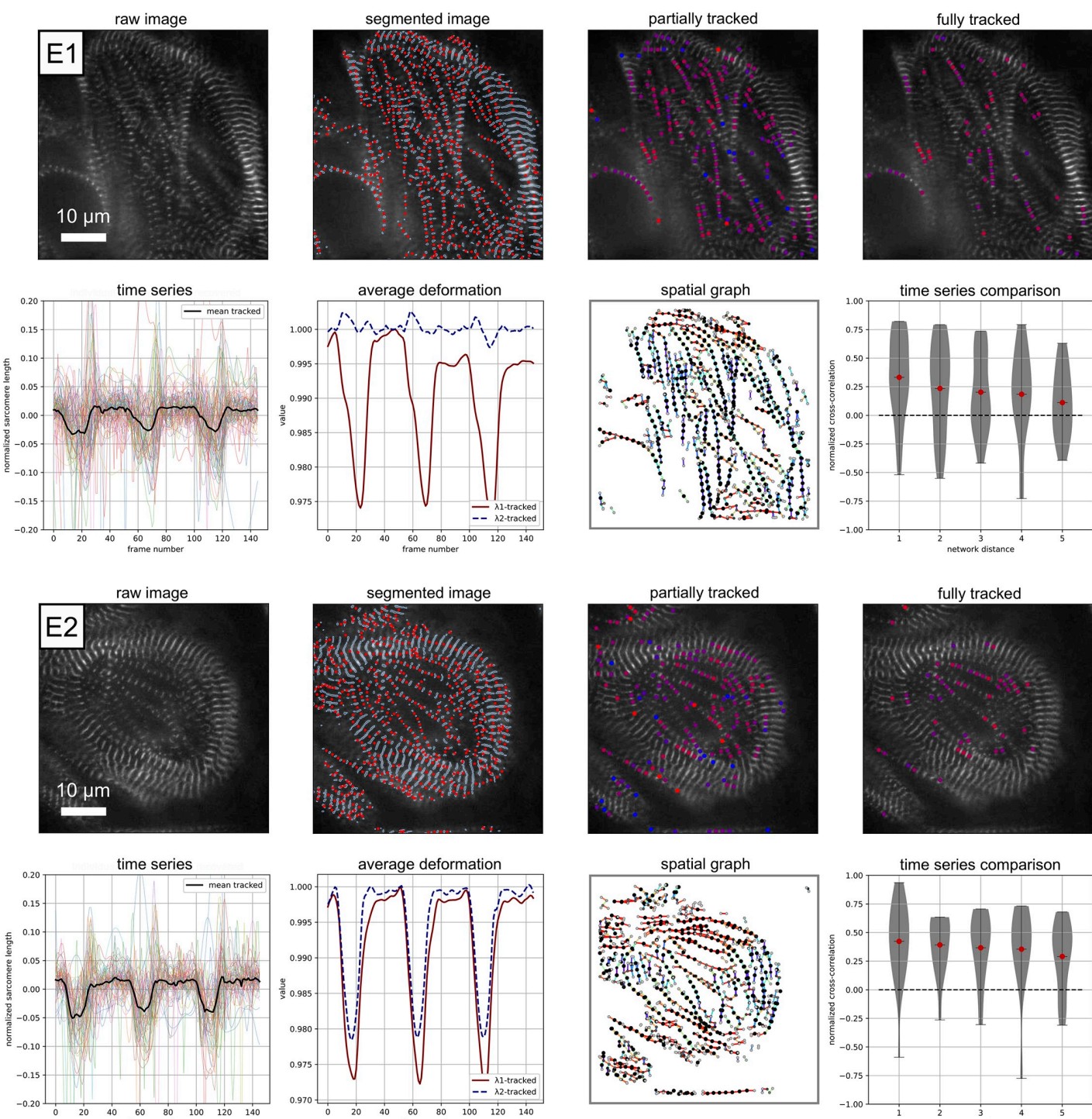

**Fig 7. Example performance on experimental data E1 and E2:** Raw image visualization; z-disc and sarcomere segmentation; Sarcomeres that are tracked for 1/3 or more of the movie, red corresponds to a contracted state, blue corresponds to a relaxed state (frame 20); Sarcomeres that are tracked for 3/4 or more of the movie, red corresponds to a contracted state, blue corresponds to a relaxed state (frame 20), note that these sarcomeres are included in the time series analysis; Individual time series data for normalized sarcomere length; Average deformation behavior; Illustration of the data represented as a spatial graph, color corresponds to sarcomere orientation; Normalized sarcomere time series cross-correlation score plotted as a function of the distance along the network. S1 and S2 Movies are included for further visualization.

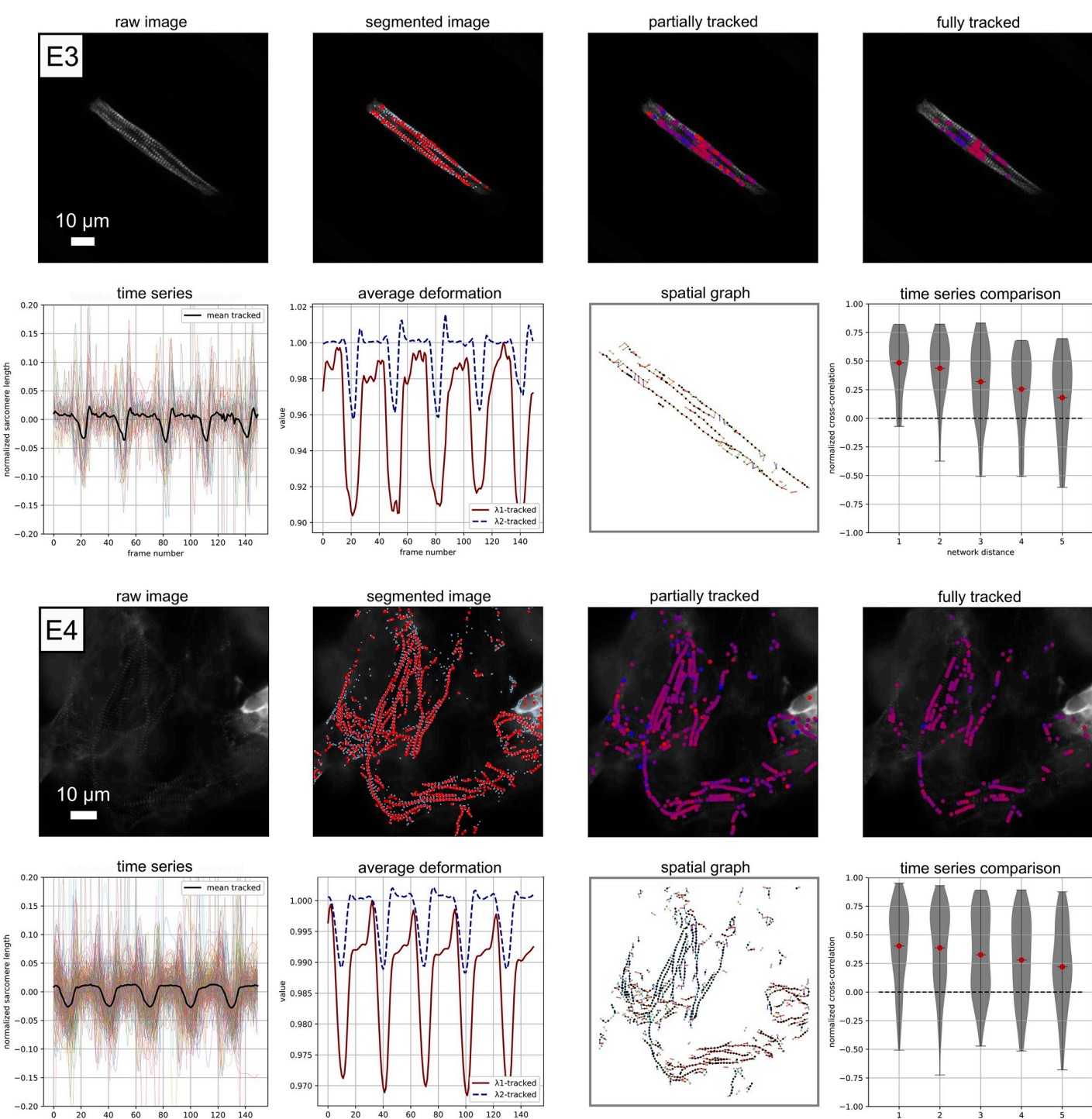

**Fig 8. Example performance on experimental data E3 and E4:** Raw image visualization; z-disc and sarcomere segmentation; Sarcomeres that are tracked for 1/3 or more of the movie, red corresponds to a contracted state, blue corresponds to a relaxed state (frame 50 for E3, frame 40 for E4); Sarcomeres that are tracked for 3/4 or more of the movie, red corresponds to a contracted state, blue corresponds to a relaxed state (frame 50 for E3, frame 40 for E4), note that these sarcomeres are included in the time series analysis; Individual time series data for normalized sarcomere length; Average deformation behavior; Illustration of the data represented as a spatial graph, color corresponds to sarcomere orientation; Normalized sarcomere time series cross-correlation score plotted as a function of the distance along the network. S3 and S4 Movies are included for further visualization.

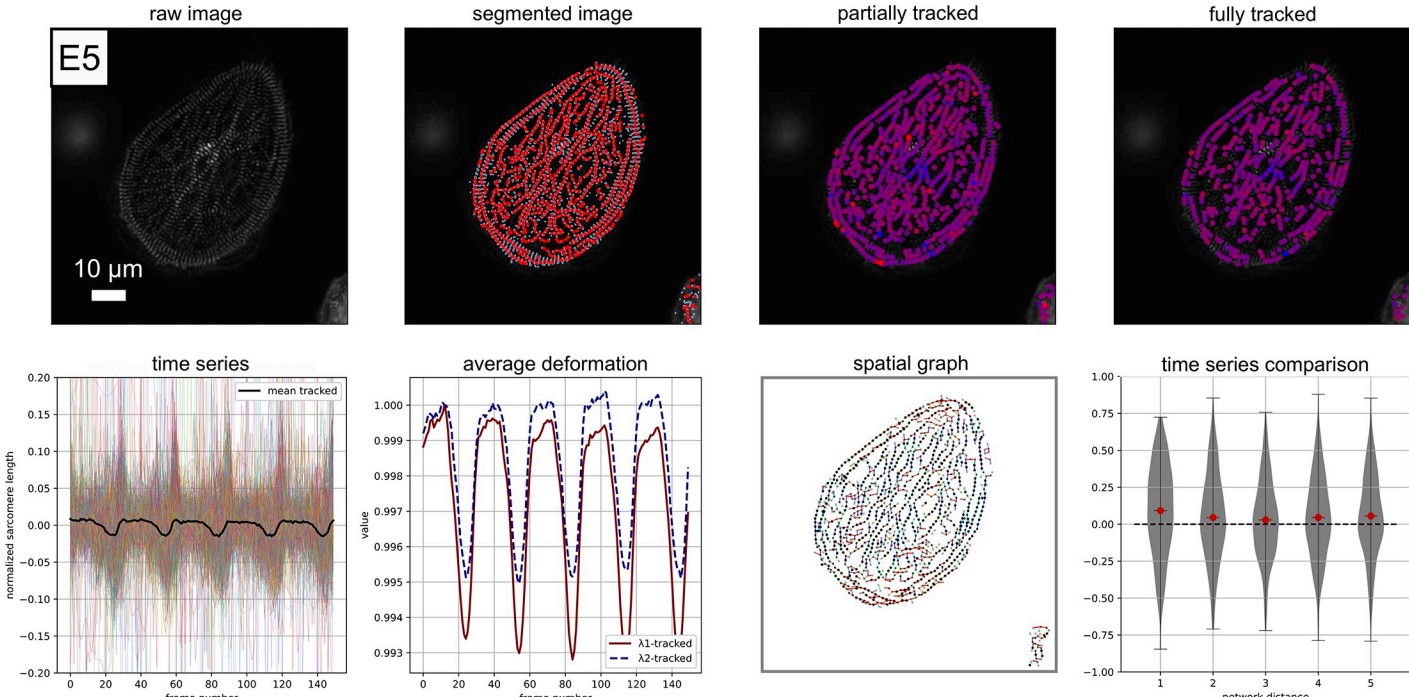

**Fig 9.** Example performance on experimental data E5: Raw image visualization; z-disc and sarcomere segmentation; Sarcomeres that are tracked for 1/3 or more of the movie, red corresponds to a contracted state, blue corresponds to a relaxed state (frame 50); Sarcomeres that are tracked for 3/4 or more of the movie, red corresponds to a contracted state, blue corresponds to a relaxed state (frame 50), note that these sarcomeres are included in the time series analysis; Individual time series data for normalized sarcomere length; Average deformation behavior; Illustration of the data represented as a spatial graph, color corresponds to sarcomere orientation; Normalized sarcomere time series cross-correlation score plotted as a function of the distance along the network. S5 Movie is included for further visualization.

In Fig 9, we show the performance of Sarc-Graph on example E5. Example E5 is of a hiPSC-CM on a glass slide with poor fiber alignment, curved fibers, and relatively low deformation over the course of the movie. With Sarc-Graph, we are able to segment approximately 800 and track 539 sarcomeres for example E5. Though individual sarcomeres in this movie contract substantially ($\tilde{s} = 0.13$), there is low overall synchrony, and overall average deformation is quite low ($C_{iso} = 0.0060$). Despite low synchrony, Sarc-Graph still detects 5 distinct contractions in both the average time series plot and the $\lambda_1^{avg}$ and $\lambda_2^{avg}$ time series plot. Notably, overall contraction is also slightly higher ($C_{\parallel} > C_{iso}$) in the dominant direction of sarcomere orientation. This example is notable because Sarc-Graph is able to segment and track a high number of sarcomeres from an irregular geometry and detect subtle sarcomere behavior beyond the limits of what has been accomplished with the previous state of the art [10].

**Table 3. Sarc-Graph computed $\tilde{s}$, $s_{avg}$, OOP, $C_{iso}$, and $C_{\parallel}$ for each experimental data example.**

|  | $\tilde{s}$ | $s_{avg}$ | OOP | $C_{iso}$ | $C_{\parallel}$ |
|---|---|---|---|---|---|
| E1 | 0.17 | 0.049 | 0.48 | 0.015 | 0.026 |
| E2 | 0.15 | 0.07 | 0.29 | 0.025 | 0.028 |
| E3 | 0.18 | 0.061 | 0.68 | 0.069 | 0.041 |
| E4 | 0.13 | 0.041 | 0.16 | 0.021 | 0.027 |
| E5 | 0.13 | 0.02 | 0.38 | 0.0060 | 0.0066 |

To complement the results of analyzing movies E1-E5, we also apply Sarc-Graph to two experimental datasets published by others [6, 12]. In both cases, we are only evaluating the segmentation functionality of the code because the datasets only contain still images. In S1 Text, we compare sarcomere length predicted by Sarc-Graph to the sarcomere length predicted by the SOTA framework [12]. In S2 Text, we demonstrate the feasibility of constructing a machine learning phenotype classification model using Sarc-Graph derived metrics with the data published with Pasqualini et al. 2015 [6]. These results help provide broader context of Sarc-Graph functionality and show the performance of Sarc-Graph on datasets that the code was not originally designed for.

### Current limitations and future directions

Based on the results from both synthetic and experimental data, it is clear that Sarc-Graph is a powerful software tool for segmentation, tracking, and analysis of sarcomeres in hiPSC-CMs. However, there are multiple current limitations and opportunities for further improvement. Key limitations are as follows:

- Potential bias in interpreting data due to variations in sarcomere width. At present, segmented sarcomeres are not weighted by size in any of the analysis steps. Therefore, a large (i.e., wide) sarcomere will have the same importance as a small (i.e., narrow) sarcomere in analysis. Because sarcomere width is measured and saved, it would be straightforward to apply a width based weighting factor during the analysis step in future work and/or compare segmented sarcomere widths between groups.

- Potential bias in segmentation due to missing poorly formed sarcomeres. As a general rule of thumb, sarcomeres (i.e., z-disc pairs) that are not discernible to the human eye will not be properly segmented by Sarc-Graph. In some cases, Sarc-Graph may segment sarcomere-like shapes, but due to inconsistency between frames these shapes will rarely be both segmented and tracked. In general, Sarc-Graph reported metrics have not been explored for poorly formed individual sarcomeres and, at this point in time, are not suitable for these applications. An example of what applying Sarc-Graph to poorly formed sarcomeres might look like is shown in S1 Text.

- Potential bias due to an inability to track all sarcomeres. As seen in both the synthetic and experimental data examples, it is typical for many more sarcomeres to be segmented than tracked. In some cases, segmented sarcomeres will be spurious and because these spurious sarcomere-like conditions are less likely to persist across frames the tracking step provides a beneficial additional filter to remove them. However, in many cases, sarcomere tracking will fail due to either large deformations or deformation with respect to the image plane. Partial tracking becomes a concern when there is bias in the type or location of sarcomeres that are tracked. To address this, there are two options: (1) specify a smaller number of frames to track (e.g. only a single contraction) thus reducing the chance of motion outside the image plane; (2) use the `compare_tracked_untracked()` function to compare the properties of tracked and untracked sarcomeres. If there are notable systematic differences, further investigation may be necessary. Future work will focus on making multiple tweaks to improve the performance of Sarc-Graph in both consistent segmentation, tracking, and subsequent statistical analysis.

- Lack of explicit force measurements. Our proposed metrics summarize kinematic information only. During cell contraction, individual sarcomere chains may bend, buckle, rearrange, and interact with other cell components that are not directly visualized [43].

Though Sarc-Graph is able to capture the kinematic aspects of bending, buckling, and re-arrangement, Sarc-Graph is unable to capture any direct information about force. Therefore, the proposed contraction metrics should be thought of strictly as kinematic contraction. In future work, Sarc-Graph metrics can be combined with additional experimental information and modeling to better determine the relationship between sarcomere length, motion, and force.

Future work will focus on both addressing these limitations and comparing the Sarc-Graph metrics to alternative approaches to capturing cell contractile function such as Traction Force Microscopy [8, 44, 45]. Critically, Sarc-Graph is a platform thorough which biologically relevant length and time scales for hiPSC-CM contraction can be uncovered. For example, Sarc-Graph can be used to understand the relationship between individual sarcomere shortening and overall cell deformation. By extracting explicitly defined relevant information from these information rich movies of hiPSC-CM contraction, Sarc-Graph will help users compare between groups of cells and help uncover fundamental relationships between external conditions, morphology, and contractile function.

## Conclusion

The objective of this work is to provide an open source computational framework to quantitatively analyze movies of beating hiPSC-CMs. To accomplish this, we introduce tools to segment and track z-discs and sarcomeres, and analyze their spatiotemporal behavior. Notably, we are able to automatically segment and track a high number of sarcomeres across multiple experimental conditions without any input parameter tuning, and we introduce two important new approaches for the analysis of beating hiPSC-CM: a method for computing average deformation gradient, and a method for treating the hiPSC-CMs as spatial graphs. Looking forward, we see this work as an important tool for substantial future study of hiPSC-CM behavior. Finally, our proposed computational framework is heavily documented and has a modular extensible design. We anticipate that many components of our open source software will be directly useful to other researchers.

## Supporting information

**S1 Text. A direct comparison to SarcOmere Texture Analysis (SOTA) software from Sutcliffe et al. [12].** Direct comparison to an alternative method for automatically quantifying morphology of single images. **Fig A. Pre-processing with a Gaussian filter**. The effect of applying a Gaussian filter as a pre-processing step on Sarc-Graph illustrated on data from Sutcliffe et al. 2018 [12]. **Fig B. Direct comparison between Sarc-Graph and SOTA** [12]. Comparison between SOTA and Sarc-Graph of sarcomere segmentation and analysis for the images shown in Figure 6 of Sutcliffe et al. 2018 [12].
(PDF)

**S2 Text. A direct comparison to Pasqualini et al. [6].** Evaluation of Sarc-Graph as a potential tool for creating features suitable for machine learning based classification. **Fig A. Pasqualini et al**. [6] **dataset**. Visualization of the un-processed dataset from Pasqualini et al. [6]. **Fig B. Pasqualini et al**. [6] **dataset segmented**. Visualization of the dataset from Pasqualini et al. [6] segmented with Sarc-Graph. **Fig C. Pasqualini et al**. [6] **dataset analysis results**. Visualization of the information obtained from the Sarc-Graph segmentation shown in Fig B in S2 Text.
(PDF)

**S3 Text. Influence of image resolution on Sarc-Graph performance.** Presentation and analysis of more synthetic data at multiple image resolutions (this is similar to Fig 5). **Fig A.**

**Demonstration of varied image resolution**. Performance of Sarc-Graph in segmentation and tracking with respect to image resolution. **Fig B. $F_{avg}$ analysis at multiple image resolutions**. Comparison of $F_{avg}$ computed with Sarc-Graph to the ground truth at different rendering resolutions. **Fig C. OOP analysis at multiple image resolutions**. Comparison of *OOP* computed with Sarc-Graph to the ground truth at different rendering resolutions. **Fig D. Angle analysis at multiple image resolutions**. Comparison of sarcomere angle computed with Sarc-Graph to the ground truth at different rendering resolutions.
(PDF)

**S4 Text. A direct comparison to SarcTrack software from Toepfer et al. [10].** Direct comparison to an alternative method for automated segmentation and tracking of hiPSC-CMs. **Fig A. SarcTrack [10] comparison 1**. Synthetic data and comparison to SarcTrack [10], example "411." **Fig B. SarcTrack [10] comparison 2**. Synthetic data and comparison to SarcTrack [10], example "412." **Fig C. SarcTrack [10] comparison 3**. Synthetic data and comparison to SarcTrack [10], example "421." **Fig D. SarcTrack [10] comparison 4**. Synthetic data and comparison to SarcTrack [10], example "422." **Fig E. SarcTrack [10] comparison 5**. Synthetic data and comparison to SarcTrack [10], example "1011." **Fig F. SarcTrack [10] comparison 6**. Synthetic data and comparison to SarcTrack [10], example "1012." **Fig G. SarcTrack [10] comparison 7**. Synthetic data and comparison to SarcTrack [10], example "1021." **Fig H. SarcTrack [10] comparison 8**. Synthetic data and comparison to SarcTrack [10], example "1022.".
(PDF)

**S1 Movie. A movie of E1.** The movie contains a schematic illustration of deformation gradient $F_{avg}$ and tracked sarcomeres overlaid (sarcomere color corresponds to contraction level with red indicating the highest level of contraction), magnitudes of average principal stretches ($\lambda_1$, $\lambda_2$ computed from $F_{avg}$), and average normalized sarcomere length, both with respect to the movie frame number. This movie is a supplement to Fig 7, sample E1.
(MP4)

**S2 Movie. A movie of E2.** The movie contains a schematic illustration of deformation gradient $F_{avg}$ and tracked sarcomeres overlaid (sarcomere color corresponds to contraction level with red indicating the highest level of contraction), magnitudes of average principal stretches ($\lambda_1$, $\lambda_2$ computed from $F_{avg}$), and average normalized sarcomere length, both with respect to the movie frame number. This movie is a supplement to Fig 7, sample E2.
(MP4)

**S3 Movie. A movie of E3.** The movie contains a schematic illustration of deformation gradient $F_{avg}$ and tracked sarcomeres overlaid (sarcomere color corresponds to contraction level with red indicating the highest level of contraction), magnitudes of average principal stretches ($\lambda_1$, $\lambda_2$ computed from $F_{avg}$), and average normalized sarcomere length, both with respect to the movie frame number. This movie is a supplement to Fig 8, sample E3.
(MP4)

**S4 Movie. A movie of E4.** The movie contains a schematic illustration of deformation gradient $F_{avg}$ and tracked sarcomeres overlaid (sarcomere color corresponds to contraction level with red indicating the highest level of contraction), magnitudes of average principal stretches ($\lambda_1$, $\lambda_2$ computed from $F_{avg}$), and average normalized sarcomere length, both with respect to the movie frame number. This movie is a supplement to Fig 8, sample E4.
(MP4)

**S5 Movie. A movie of E5.** The movie contains a schematic illustration of deformation gradient $F_{avg}$ and tracked sarcomeres overlaid (sarcomere color corresponds to contraction level with

red indicating the highest level of contraction), magnitudes of average principal stretches ($\lambda_1$, $\lambda_2$ computed from $\mathbf{F}_{avg}$), and average normalized sarcomere length, both with respect to the movie frame number. This movie is a supplement to Fig 9, sample E5. A still frame from this movie is also shown in Fig 1.
(MP4)

## Author Contributions

**Conceptualization:** Kehan Zhang, Christopher S. Chen, Emma Lejeune.

**Data curation:** Kehan Zhang, Emma Lejeune.

**Funding acquisition:** Christopher S. Chen, Emma Lejeune.

**Investigation:** Bill Zhao.

**Methodology:** Emma Lejeune.

**Resources:** Christopher S. Chen, Emma Lejeune.

**Software:** Emma Lejeune.

**Supervision:** Christopher S. Chen, Emma Lejeune.

**Validation:** Emma Lejeune.

**Visualization:** Emma Lejeune.

**Writing – original draft:** Bill Zhao, Kehan Zhang, Emma Lejeune.

**Writing – review & editing:** Christopher S. Chen, Emma Lejeune.

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
