## [Decision Letter · Decision Letter 0]

21 Mar 2021

Dear Dr. Lejeune,

Thank you very much for submitting your manuscript "Sarc-Graph: Automated segmentation, tracking, and analysis of sarcomeres in hiPSC-derived cardiomyocytes" for consideration at PLOS Computational Biology.

As with all papers reviewed by the journal, your manuscript was reviewed by members of the editorial board and by several independent reviewers. In light of the reviews (below this email), we would like to invite the resubmission of a significantly-revised version that takes into account the reviewers' comments.

We cannot make any decision about publication until we have seen the revised manuscript and your response to the reviewers' comments. Your revised manuscript is also likely to be sent to reviewers for further evaluation.

Sincerely,

Mihaela Pertea

Software Editor

PLOS Computational Biology

Mihaela Pertea

Software Editor

PLOS Computational Biology

Reviewer's Responses to Questions

**Comments to the Authors:**

Reviewer #1: This manuscript describes a method to automatically track sarcomeres in hiPSC derived cardiomyocytes. This is an important problem that has been the subject of multiple publications lately as it is imperative that stem-cell derived cardiomyocytes are compared without bias among test populations. The method uses segmentation to track sarcomeric architecture through time and the outputs include metrics that quantify a measure of contractility, sarcomere shortening, and others. While this article addresses an important issue in the field, there are some major issues that need to be addressed prior to proper review:

Major:

1. The introduction reads like a discussion section in that it presents the results of the paper instead of the background. It would be more appropriate to more fully describe the previous computational frameworks cited in the paper. The contrasting with the present work is more appropriate for the discussion section.

2. There is a logical conflict between the stated goals of the paper in the introduction and the approach taken. Indeed, as stated, there is a pressing need in the field to analyze iPSC-derived cardiomyocyte structure. However, this is a challenging problem because the sarcomeres in these cells are often not fully formed (i.e. immature as mentioned in the introduction). However, all of the synthetic data presented in the paper assumes perfectly formed sarcomeres. Further, the images picked for the optimization/validation of this method are cherry picked as very good single cell iPSC-derived cardiomyocytes except for those presented in Figure 8-E4. However, the latter is not at a high enough resolution to judge if the reconstruction of the architecture is of good quality. In general, this is a non-trivial requirement for an automated analysis code, but the authors need to discuss how ignoring immature sarcomeres affects their results.

3. The other issue with iPSC derived cardiomyocytes is that the sarcomeres will often deform out of sync with each other. It is not clear how the metrics described in Eq. 1-4 would be affected by non-synchronous dynamics. It is imperative that such motion be considered theoretically in order to interpret experimental results.

4. As this is a paper targeted at the computational/image analysis community, there needs to be more context in the description of the experimental results. Are these typical cells? Are these the best possible cells achieved with the given technique? Would the image analysis work on iPSC-derived cardiac tissues that were not well formed? Basically the reader needs more context to judge how challenged the code was.

5. Additionally, the brevity of the experimental section was a bit concerning if any of the data was new. If all of the experimental data has already been published previously, that needs to be more clearly stated in each row of Table 1 where the published data can be found. As I read it, there need to be more details for E3-E5.

6. Generally, the paper organization needs to be improved. Some of the descriptions in the “Materials and Methods” section should be moved into the “Results” Section. In particular, “Computing average deformation” and “Comparing average deformation to average sarcomere shortening and orientational order parameter” would be better suited for the “Results” section.

7. There looks to be a huge loss in sarcomeres that are partially tracked vs fully tracked (Fig. 7-8). Additionally, there is a huge loss in sarcomeres between the segmented image and the partially tracked image. Based on Figure 7, it appears that sarcomeres from parallel myofibrils that are aligned or in register, are treated as one sarcomere and are consequently not partially or fully tracked. This needs to be explained and put into context of applicability of this code.

8. It would also improve the paper to directly compare the segmentation and metrics reported by “Sarc-Graph” with those done by other software (Pasqualini et al., 2015; “SarcOptiM” by Pasqualin et al., 2016; “SarcOmere Texture Analysis” Sutcliffe et al., 2018; “SarcTrack” by Toepher et al., 2019). In particular, the segmentation and tracking should be directly compared with “SarcTrack” by Toepher et al., 2019. In that work they provide metrics that quantify contractility and sarcomere structure, such as duration of contraction and relaxation, sarcomere shortening, and sarcomere length. Will computing the sarcomere shortening with their software yield the same results as Sarc-Graph? Why or why not? Further, there should be more discussion on how the contractility metrics reported by Sarc-Graph (Favg, Ciso, C||) relate to other contractility metrics or any methods that directly measure contractility (Traction Force Microscopy).

9. Further to the above point, it is unclear how the authors differentiate in their measure of contractility between the stress generated by shortening of the sarcomeres, the possible bending of the sarcomeres as the myofibril is deforming, and the resultant applied force. This is a very challenging problem as the material properties of the myofibril are non-simplistic. As a result, the metrics proposed in this work are at best an indirect measure of contractility. As such, it is important that the authors clearly state the actual physical meaning of the proposed metrics. Without such a clear-cut explanation, these would be greatly misused and misinterpreted in the field.

10. The Discussion section lacks a serious comparison to other works, existing metrics, and a in depth discussion of how the metrics should be applied AND the situations where the metrics should not be used.

11. Further, the discussion section should be expanded to explain how this new method contributes to a deeper understanding of the biology of the IPSC-derived cardiomyocytes.

12. For this work to be impactful better documentation is needed. In particular, setup.py / requirements.txt file should be added and it should include better instructions on how to set up the environment, including package versions. The package links included in the README.md were not sufficient for this reviewer to be able to run Sarc-Graph. This is an important step to be able to properly review this manuscript.

Minor:

1. Z-discs is spelled “z-disk” in Fig 3c.

2. Experimental movies/images should have scale bars.

3. It would be helpful if the sub-figures actually had labels that were referenced specifically in the text.

In summary, while this paper presents a possible solution to a pressing problem in the field, its results were not presented in a manner that would allow for in depth analysis and interpretation of the applicability of the code to a wide range of experimental data.

Reviewer #2: "Sarc-Graph: Automated segmentation, tracking, and analysis of sarcomeres in hiPSC-derived cardiomyocytes" is a particularly interesting manuscript regarding an automated segmentation and mechanical analysis procedure for sarcomeres in cardiomyocytes. Broadly, the image processing procedures are well-executed and characterized, and I am appreciative of the authors' attention to detail in their assessment of the robustness of their algorithms and analysis. There are a few areas where the paper can be shored up and perhaps made more intuitive to a broad (i.e. non-cardiomyocyte-related) audience, but I'm quite supportive of the work.

I'll begin with some broad points, and follow with specific line items.

A1) Some more detail about the structure and optical imaging of sarcomeres would be welcome. Perhaps a schematic with some definitions would clarify to the uninformed what a sarcomere, z-disc, etc. are considered in terms of lengths, bounds, etc.

A2) There are a variety of typos, one of the more common ones was permuting the ‘u’ and ‘o’ in fluorescent. It’d be worth an additional check to catch these, some hyphenation issues, etc.

A3) Regarding angular error: the pixel error from z-discs being small may have an outsized effect on the measured angular rotation between neighboring discs. Can this be approximated? What effects does this propagate to F? This would be of interest for imaging with different objectives, etc. There’s allusion to this in ln.207, but even an approximate quantification would be helpful.

A4) The metric of “shortening” is somewhat opaque to me, possibly just because of the +1 in the denominator. Why is it defined as it is, rather than e.g. simply delta-y/y_max? Why not use something conventional, like axial strain?

A5) Is there any physical interpretation of the structural tensor T? I’d also recommend using a different set of variables for the eigenvalues, as the eigenvectors for the stretch tensor are given as u already. Perhaps change eig(U) to lambda_i (e.g. as in Holtzapfel).

A6) The use of the Jacobian of the deformation gradient tensor seems incorrect to me. J is conventionally det(F), and is defined as the norm of F in ln.397. Regarding C_iso as well, it seems to me that the square root would be true for an assumed 2D setup, where in the 3D version it should be a cube root instead.

A7) I just want to reiterate that the tests for robustness are really well done, and I especially support the use of the viridis map and matplotlib package for accessibility. Nice work.

Minor comments by line:

Ln.9. Unsure what “more pressing” is being compared to

25. Shown

Table 1. What direction is vertical?

Fig. 3 legend. Is the “raw” 2D input image experimental or synthetic?

122. Fluctuating

139. Its

176. What’s considered a particle?

249. Sarcomeres is pluralized, possibly shouldn’t be

362. Two ns in on

**Have all data underlying the figures and results presented in the manuscript been provided?**

Reviewer #1: Yes

Reviewer #2: Yes

PLOS authors have the option to publish the peer review history of their article (what does this mean?). If published, this will include your full peer review and any attached files.

Reviewer #1: No

Reviewer #2: No
---

## [Decision Letter · Decision Letter 1]

30 May 2021

Dear Dr. Lejeune,

Thank you very much for submitting your manuscript "Sarc-Graph: Automated segmentation, tracking, and analysis of sarcomeres in hiPSC-derived cardiomyocytes" for consideration at PLOS Computational Biology. As with all papers reviewed by the journal, your manuscript was reviewed by members of the editorial board and by several independent reviewers. The reviewers appreciated the attention to an important topic. Based on the reviews, we are likely to accept this manuscript for publication, providing that you modify the manuscript according to the review recommendations.

Please, can you test your software on multiple environments and address the error that one of the reviewers is getting while running the software? 

Sincerely,

Mihaela Pertea

Software Editor

PLOS Computational Biology

Mihaela Pertea

Software Editor

PLOS Computational Biology

[LINK]

Reviewer's Responses to Questions

**Comments to the Authors:**

Reviewer #1: The authors have addressed almost all of my concerns. However, I was still unable to run the code. It give the following error:

Traceback (most recent call last):

File "run_code.py", line 1, in <module>

import file_pre_processing as fpp

File "/mnt/c/Users/Admin/Documents/Python/Sarc-Graph/file_pre_processing.py", line 1, in <module>

import av

ModuleNotFoundError: No module named 'av'

It is essential that the code can be run by the readers for this work to have the desired impact.</module></module>

Reviewer #2: I'm grateful for your attention to detail in the improved version of this manuscript, and enthusiastically support its acceptance to PLoS Comp Biol.

**Have the authors made all data and (if applicable) computational code underlying the findings in their manuscript fully available?**

Reviewer #1: Yes

Reviewer #2: None

PLOS authors have the option to publish the peer review history of their article (what does this mean?). If published, this will include your full peer review and any attached files.

Reviewer #1: No

Reviewer #2: No

Figure Files:

Data Requirements:

Reproducibility:

References:

---

## [Decision Letter · Decision Letter 2]

10 Sep 2021

Dear Dr. Lejeune,

We are pleased to inform you that your manuscript 'Sarc-Graph: Automated segmentation, tracking, and analysis of sarcomeres in hiPSC-derived cardiomyocytes' has been provisionally accepted for publication in PLOS Computational Biology.

Best regards,

Mihaela Pertea

Software Editor

PLOS Computational Biology

Feilim Mac Gabhann

Editor-in-Chief

PLOS Computational Biology

Reviewer's Responses to Questions

**Comments to the Authors:**

Reviewer #1: The authors have addressed all of my concerns.

**Have the authors made all data and (if applicable) computational code underlying the findings in their manuscript fully available?**

Reviewer #1: Yes

PLOS authors have the option to publish the peer review history of their article (what does this mean?). If published, this will include your full peer review and any attached files.

Reviewer #1: No

---

## [Editor Report · Acceptance letter]

30 Sep 2021

PCOMPBIOL-D-21-00225R2 

Sarc-Graph: Automated segmentation, tracking, and analysis of sarcomeres in hiPSC-derived cardiomyocytes

Dear Dr Lejeune,

I am pleased to inform you that your manuscript has been formally accepted for publication in PLOS Computational Biology. Your manuscript is now with our production department and you will be notified of the publication date in due course.

With kind regards,

Olena Szabo
